# SUITOR: Selecting the number of mutational signatures through cross-validation

**Donghyuk Lee[1], Difei Wang[2], Xiaohong R. Yang[2], Jianxin Shi[2], Maria Teresa Landi[2], Bin Zhu[2]***

**1** Department of Statistics, Pusan National University, Busan, Korea, **2** Division of Cancer Epidemiology and Genetics, National Cancer Institute, National Institutes of Health, Bethesda, Maryland, United States of America

* bin.zhu@nih.gov

## Abstract

For *de novo* mutational signature analysis, the critical first step is to decide how many signatures should be expected in a cancer genomics study. An incorrect number could mislead downstream analyses. Here we present SUITOR (Selecting the nUmber of mutatIonal signaTures thrOugh cRoss-validation), an unsupervised cross-validation method that requires little assumptions and no numerical approximations to select the optimal number of signatures without overfitting the data. *In vitro* studies and *in silico* simulations demonstrated that SUITOR can correctly identify signatures, some of which were missed by other widely used methods. Applied to 2,540 whole-genome sequenced tumors across 22 cancer types, SUITOR selected signatures with the smallest prediction errors and almost all signatures of breast cancer selected by SUITOR were validated in an independent breast cancer study. SUITOR is a powerful tool to select the optimal number of mutational signatures, facilitating downstream analyses with etiological or therapeutic importance.

**Data Availability Statement:** All relevant data are within the manuscript and its Supporting Information files. SUITOR: https://github.com/binzhulab/SUITOR In vitro studies http://medgen.medschl.cam.ac.uk/serena-nik-zainal/ Sanger

## Author summary

Mutational signatures are the footprints of exogenous exposures and endogenous mutational processes on the cancer genomes. To estimate *de novo* mutational signatures, the first step is to decide how many signatures should be extracted in a cancer genomics study, which determines downstream analytical steps and has been insufficiently studied. We developed SUITOR, an unsupervised cross-validation method to select the optimal number of signatures without overfitting the data. We demonstrated SUITOR's superior performance using *in vitro* experimental studies, *in silico* simulations and *in vivo* pan-cancer applications of 2,540 whole-genome sequenced tumors across 22 cancer types, and validated signatures of breast cancer in additional 440 breast tumors. SUITOR advances the methodological frontier of identifying *de novo* mutational signatures and would help discover the causes of cancer and the means of cancer prevention and treatment.

whole genome sequencing breast cancer study ftp://ftp.sanger.ac.uk/pub/cancer/Nik-ZainalEtAl-560BreastGenomes (Note: it's an FTP site. For Mac OS, please use the Finder window to access it; for Windows, use the File Explorer in Windows 8, or Windows Explorer in previous versions; the FTP site can also be accessed by FTP Clients, such as FileZilla, WinSCP and CyberDuck.) The Pan-Cancer Analysis of Whole Genomes (PCAWG) study https://www.synapse.org/#!Synapse:syn11726620.

**Funding:** This research was supported by the Intramural Research Program of the National Institutes of Health, National Cancer Institute, Division of Cancer Epidemiology and Genetics (DCEG). The funders had no role in study design, data collection and analysis, decision to publish, or preparation of the manuscript.

**Competing interests:** The authors have declared that no competing interests exist.

This is a *PLOS Computational Biology* Methods paper.

## Background

Mutational signatures are patterns of somatic mutations imprinted on the cancer genome by operative mutational processes, including signatures of single base substitution [1], doublet base substitutions [1], structural variations [2,3] and copy number alterations [4–6]. For example, seventy-eight single base substitution mutational signatures have been identified across cancer types (https://cancer.sanger.ac.uk/signatures/) [1,7], with some associated with exogenous mutagenic exposures [8–10] and endogenous mutational processes [11–14]. Moreover, mutational signatures have been associated with cancer predisposition genes (e.g., *NTHL1* in multiple cancer types [15], including breast cancer [16]), and used to stratify cancer patients [17–20] for precision treatment. In these studies, deciding the expected number of signatures is the pivotal first step, which determines the downstream steps of extracting signature profiles, estimating signature activities and stratifying tumors based on signatures for treatment. As an example, the Pan-Cancer Analysis of Whole Genomes (PCAWG) consortium reported that the discordance between the extracted and known signatures is usually caused by the difficulty in selecting the correct number of signatures [1].

There are two main types of mutational signatures analysis [21–23]: signature extraction and signature refitting. Signature extraction aims to extract *de novo* signature profiles [24–28] while signature refitting to estimate signature activities based on reference mutational signatures with potential applicability in the clinical setting [29,30]. Our interest is on selecting the correct number of *de novo* mutational signatures for signature extraction in cancer genomics studies [31,32], which has been insufficiently explored. SomaticSignatures [24] measures the goodness of fit of the number of signatures based on the residual sum of squares and the explained variance with no automatic selection criterion. SigProfilerExtractor [25] considers the mean reconstruction error and the stability of signature extraction; however, it is unclear how these features could be combined to jointly predict the number of signatures. EMu [26] and signeR [27] adopt a Bayesian information criterion (BIC) [33]. Although BIC is a popular model selection criterion for supervised learning (e.g., regression and classification) where the number of parameters is fixed, it may not be applicable to unsupervised learning, including mutational signature analysis, where the number of parameters increases with the sample size (other limitations of BIC are elaborated in Supplementary Note 1 in S1 Text). SignatureAnalyzer [1] uses an automatic relevance determination (ARD) prior [34] which imposes a sparsity assumption on mutation profiles and activities. The number of signatures chosen by SignatureAnalyzer is sensitive to the pre-specified sparsity assumption, especially hyperparameters of the ARD prior and the tolerance level.

To overcome the limitations of previous methods, we propose selecting the number of signatures through cross-validation. Selecting the number of signatures is essentially a problem of model selection, which has been addressed by cross-validation in other research areas [35,36], including identification of cancer subtypes [37], exploration of population structure [38] and prediction of lymph node metastasis [39]. In the setting of mutational signature analysis, cross-validation splits the full dataset (here, the mutation counts) into a training set and a validation set; for a given number of signatures, these signatures are estimated in the training set and then they are used to predict the mutations in the validation set. Multiple candidate numbers of signatures are considered; and the number of signatures which predicts most closely the mutations in the validation (not the training) set is selected. Hence, cross-validation can prevent selecting too few or too many signatures (corresponding to an underfitting or

overfitting model), both of which would predict mutation counts in the validation set poorly. In addition, unlike the BIC or the ARD prior, cross-validation requires little assumptions and no numerical approximations [36]. Therefore, cross-validation provides a viable solution for selecting the correct number of signatures.

Despite being conceptually appealing, the standard cross-validation approach does not work for unsupervised mutational signature analysis. In the standard cross-validation scheme for the supervised learning (e.g., regression or classification in machine learning), it is feasible to remove a subset of subjects all together as a validation set. In contrast, cross-validation for mutational signature analysis requires retaining all tumors in the training set but removing some mutation counts from each tumor as a validation set. Consequently, missing data emerge in the training set and current methods for mutational signature analysis are inapplicable for cross-validation.

These limitations are overcome by SUITOR (Selecting the nUmber of mutatIonal signa-Tures thrOugh cRoss-validation), an unsupervised cross-validation method that selects the optimal number of signatures to attain the minimal prediction error in the validation set. SUITOR extends the probabilistic model to allow missing data in the training set, which makes cross-validation feasible. Moreover, we propose an expectation/conditional maximization (ECM) algorithm [40] to extract signature profiles, estimate signature activities and impute the missing data simultaneously. We demonstrated SUITOR's superior performance using *in vitro* experimental data, *in silico* simulations, *in vivo* applications to 2,540 tumors across 22 cancer types, and validation of signatures of breast cancer in additional 440 breast tumors. Recently, other cross-validation methods have been proposed to select the number of signatures; compared to SUITOR, CV2K focuses on selecting the number of signatures only (without extracting signature profiles and estimating signature activities) and is based on random, not balanced separation [41]; SparseSignatures applies cross-validation to select both the number of signatures and the shrinkage parameter simultaneously, which is computationally intensive and tends to infer signatures with spiking profiles [42].

## Results

### Overview of SUITOR

Mutational signature analysis decomposes the somatic mutation type matrix. Take single base substitution (SBS) as an example. A somatic mutation type matrix **V** of size 96×N contains mutation counts for N cancer genomes and 96 SBS types. Each SBS type refers to a mutated pyrimidine (C or T) in the center and two unmutated adjacent nucleotides (flanking 5' and 3' bases) with total 4×6×4 = 96 types. For example, a genomic sequence ACG in the normal tissue is mutated to AGG in the tumor tissue. This SBS belongs to the A[C > G]G mutation type.

SUITOR is built upon a probabilistic model [43,44], for which the maximum likelihood estimation (MLE) is equivalent to the solution of non-negative matrix factorization (NMF), the most popular method for mutational signature analysis [25]. More details are included in Methods section. NMF or equivalently the Poisson NMF model requires a given number of signatures *r*, which is unknown in practice; and SUITOR can select this number empirically. The steps of SUITOR are outlined as follows (with a schematic illustration in Fig 1): 1) the mutation type matrix is separated into the training and validation sets. The training set contains missing data held out as validation data; 2) the missing data are imputed in the initial step; 3) the ECM algorithm iteratively imputes the missing data in the expectation step (E-step) and estimates the signature activities (of the **H** matrix) and profiles (of the **W** matrix) in the conditional maximization steps (CM-steps) until the ECM algorithm allows **V**≈**WH** for a given number of signatures; 4) the missing data are imputed and compared to the validation

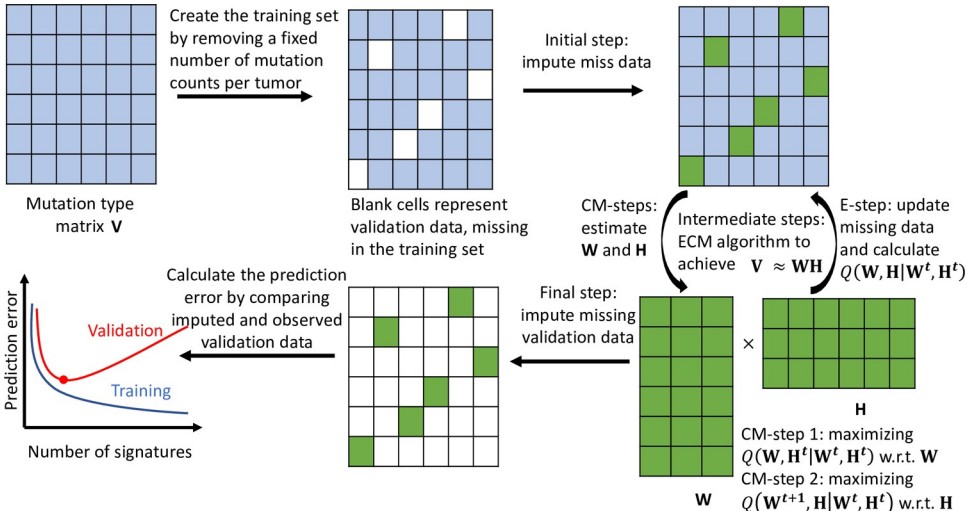

**Fig 1. A schematic overview of SUITOR.** This schematic diagram illustrates how SUITOR selects the number of *de novo* mutational signatures. Details are given in Results. Each column of the mutation type matrix *V* represents a tumor, each row a mutation type. The bottom left shows prediction error curves in the training set (blue) and validation set (red), which are manually drawn for the illustration purpose, with the red dot representing the minimal prediction error in the validation set. ECM algorithm: expectation/conditional maximization algorithm; CM-steps: conditional maximization steps; E-step: expectation step; **W**: signature profile matrix of size 96×*r*,*r* the number of signatures; **H**: signature activity matrix of size *r*×N, N the number of tumor; $Q(W,H|W^t,H^t)$: conditional expectation of complete likelihood of **W** and **H** at the (*t*+1)-th step, given estimated $W^t$ and $H^t$ in the *t*-th step of ECM algorithm.

data to calculate the prediction error in the validation set; 5) the above described steps are conducted for multiple candidate numbers of signatures and the one with the minimal prediction error will be chosen as the optimal number of signatures (corresponding to the red dot in the prediction error curve of validation set in Fig 1).

There are three key contributions of SUITOR. First, SUITOR selects the number of signatures with the minimal prediction error, which prevents selecting too few or too many signatures (as model underfitting or overfitting). Although the prediction error in the training set is reduced with increasing signatures (as illustrated by the prediction error curve of the training set in Fig 1), the prediction error in the validation set will decrease first (due to the model underfitting with insufficient signatures) and then inflate (due to the model overfitting with redundant signatures). This is the well-known bias-variance tradeoff for model complexity [45] measured by the number of signatures in the setting of mutational signature analysis. Second, the cross-validation scheme of SUITOR guarantees that the missing data pattern does not depend on the remaining or missing mutation counts in the training set. Hence, the missing data mechanism is missing completely at random (MCAR), which ensures that the estimated signature profiles and activities would not be biased due to the missing data [46]. Third, the proposed ECM algorithm for SUITOR enjoys the convergence property, which guarantees the increase of the likelihood function over iterations until the ECM algorithm converges [40].

## Evaluation of SUITOR in two *in vitro* studies

We assessed the performance of SUITOR in two experimental studies [8,11], for which the true number and profile of signatures were generated experimentally and validated *in vitro* (see details in Methods). The first study created endogenous mutational signatures through CRISPR-Cas9-mediated knockouts of DNA repair genes in an isogenic human cell line [11]. The second study generated exogenous mutational signatures in human-induced pluripotent

stem cell (iPSC) lines exposed to environmental or therapeutic mutagens [8]. For both studies, we evaluated whether SUITOR could correctly select the number of signatures and recover the profiles of single base substitution signatures. We then compared SUITOR's performance with SigProfilerExtractor, SignatureAnalyzer and signeR.

We first evaluated the *in vitro* CRISPR-Cas9-mediated knockout study of the DNA repair gene *MSH6*, which was known to induce a detectable signature [11] and hence could be used for evaluation as the positive control. SUITOR correctly detected the background signature, which existed before the knockout of *MSH6* and *MSH6* knockout-induced signature (Fig 2A) and recovered the corresponding signature profiles (Fig 2B and Table A in S1 Table). SigProfilerExtractor, SignatureAnalyzer and signeR correctly identified these two signatures as well (S1 Fig and Table A in S1 Table). Next, we extracted signatures of knockout studies of six DNA repair genes (*CHEK2*, *NEIL1*, *NUDT1*, *POLB*, *POLE* and *POLM)*, which did not induce experimentally detectable signatures [11] and hence could be used as the negative controls. SUITOR and the other three methods correctly identified one background signature only without false detection of knockout-induced signatures (S2 Fig). We conclude that in this *in vitro* study with at most two signatures, all four methods perform equally well. This is not the case when the number of signatures increases as shown below.

For the *in vitro* study of 79 exogenous mutagens, stable mutational signatures were experimentally identified for 28 mutagens. These 28 signatures are not distinct as some signatures are very similar to each other (details in Methods); for example, both benzo[a]pyrene (BaP) and benzo[a]pyrene-7,8-dihydrodiol-9,10-epoxide (BPDE) are polycyclic aromatic hydrocarbons (PAHs) which are the established mutagens in tobacco smoke (S3 Fig). Although all four methods could find the background mutational signature, SUITOR detected 9 additional signatures induced by mutagens (Fig 2C, 10 signatures, including one background signature and 9 mutagen-induced signatures), SigProfilerExtractor detected 4, SignatureAnalyzer 8, and signeR 5 additional signatures (Figs 2D and S4 and Table B in S1 Table). Next, we extracted *de novo* signature profiles and compared them with signature profiles reported in the original *in vitro* study [8]. Indeed, de *novo* signatures extracted by SUITOR, SigProfilerExtractor, SignatureAnalyzer and signeR could be matched to 10, 5, 9, 6 of 11 *in vitro* true signatures respectively with the cosine similarity threshold 0.8 (sensitivity = 91%, 45%, 82%, 55%, Table B in S1 Table). The signature of 1,2-dimethylhydrazine (1,2-DMH) was detected by SUITOR and missed by the other methods. When cosine similarity threshold was increased to 0.9, the numbers of matched signatures were reduced to 7, 5, 7, 6 for SUITOR, SigProfilerExtractor, SignatureAnalyzer and signeR, respectively (sensitivity = 64%, 45%, 64%, 55%, Table B in S1 Table). Besides detecting more true signatures, SUITOR also achieved the lowest prediction error (SUITOR:1292.5; SigProfilerExtractor:2295.4; SignatureAnalyzer:2176.6; signeR:1585.1). This indicates that the signatures detected by SUITOR in the training set predicted most closely the mutation counts in the validation set. Finally, we examined the signature activity matrices which were estimated by each method. Since each tumor subclone was exposed to one exogenous mutagen, we expected that the signature activities of the corresponding mutagen and the background signature should be high, while the activities of other signatures should be close to zero (but not necessarily as exactly zeros); consequently, when we clustered tumor subclones based on signature activities, tumor subclones exposed to the same mutagen would be clustered together and separated from others. We found that SUITOR and SignatureAnalyzer were able to separate most subclones into distinct clusters, each corresponding to a unique mutagen exposure (S5 Fig). In contrast, SigProfilerExtractor merged subclones exposed to BPDE and dibenz[a,h]anthracene diol-epoxide (DBADE); signeR mixed subclones exposed to N-ethyl-N-nitrosourea (ENU) and 1,2-DMH; and both SigProfilerExtractor and signeR mixed subclones exposed to 9-Nitrochrysense and aristolochic acid I (AAI).

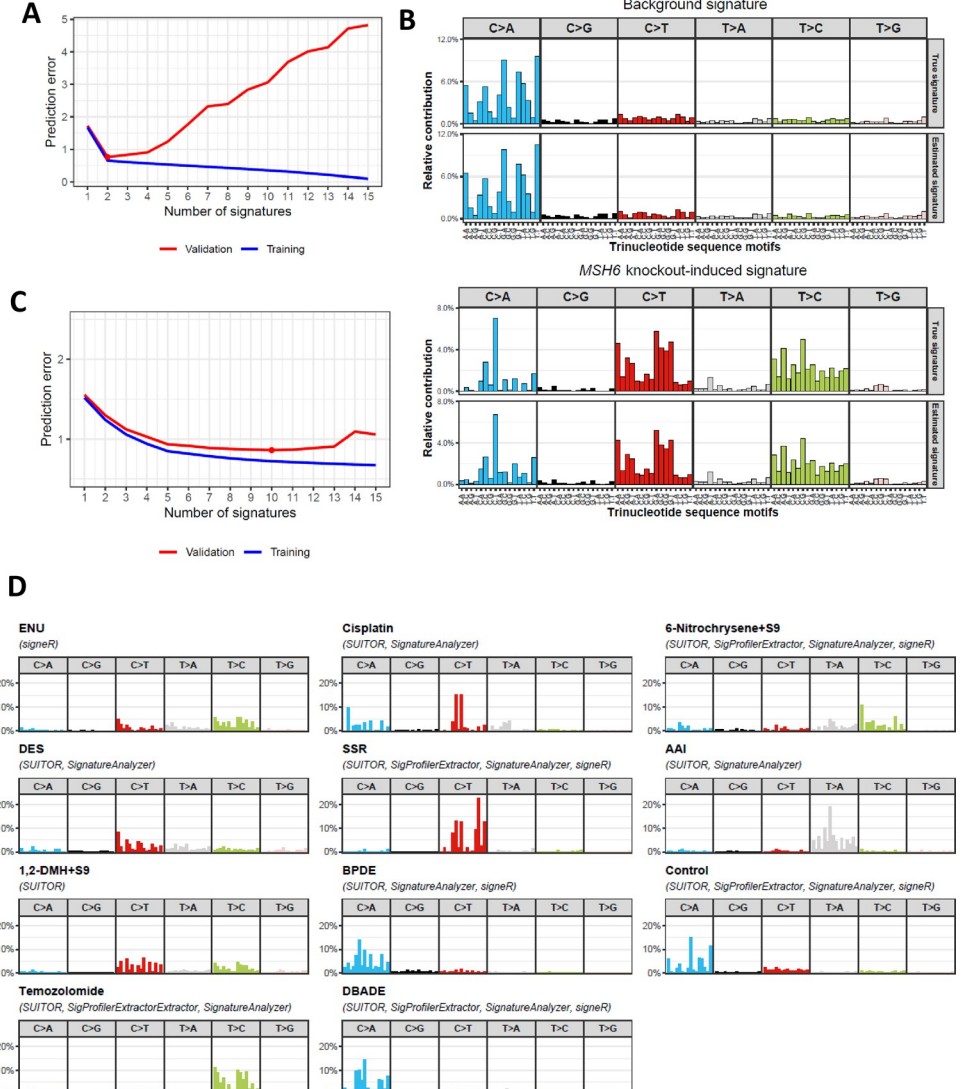

**Fig 2. *In vitro* evaluation of SUITOR and other methods.** a) Prediction errors of SUITOR for the training and validation sets of *in vitro* knockout study of DNA repair gene *MSH6*. The red dot denotes the number of signatures with the minimal prediction error in the validation set. b) Profiles of single base substitution signature estimated by SUITOR in *MSH6* gene knockout study. The x-axis indicates the 5' and 3' nucleotides for each substitution type (e.g., T [C>A]C, cytosine to adenine substitution with 5' thymine and 3' cytosine). The top panel: the true and estimated background signatures (cosine similarity = 0.991; cosine similarity = 1 suggests two profiles being identical.); the bottom panel: the true and estimated *MSH6* knockout-induced signatures (cosine similarity = 0.997). c) Prediction errors by SUITOR for the training and validation sets of *in vitro* study of environmental or therapeutic mutagens. The red dot indicates the minimal prediction error in the validation set achieved by ten signatures, including one background signature and nine mutagen-induced signatures. d) Signatures discovered from the *in vitro* study of environmental or therapeutic mutagens by four methods. Methods which could identify a given mutagen are enclosed in parentheses. ENU: N-ethyl-N-nitrosourea; 6-Nitrochrysene+S9: 6-Nitrochrysene mixed with S9 rodent liver-derived metabolic enzyme; DES: diethyl sulfate; SSR: simulated solar radiation; AAI: aristolochic acid I; 1,2-DMH+S9: 1,2-dimethylhydrazine mixed with S9 rodent liver-derived metabolic enzyme; BPDE: benzo[a]pyrene-7,8-dihydrodiol-9,10-epoxide; DBADE: dibenz[a,h]anthracene diol-epoxide.

We note that *in vitro* study of exogenous mutagens favors the method of SignatureAnalyzer. SignatureAnalyzer implicitly assumes few signatures are present per sample and hence the loadings of signatures are sparse, which holds here since each sample was treated with a single

mutagen in this *in vitro* study of exogenous mutagens. In spite of that, SUITOR performed better than SignatureAnalyzer. When the sparsity assumption does not hold, as demonstrated in *in silico* simulations and the PCAWG study described below, SignatureAnalyzer would find more false-positive signatures than the other methods. In contrast, SUITOR, which does not rely on the sparsity assumption, is not susceptible to it.

### *In silico* simulation studies

To benchmark the performance of SUITOR and other methods, we evaluated them through additional *in silico* simulations, for which the estimated signature profiles are compared to the given signature profiles as the ground truth. First, we considered a simple setting of one true signature, for which any additional signature found would be a false positive (details in Methods). SUITOR, SigProfilerExtractor and signeR correctly identified the single true signature (S6 Fig) in 20 of 20 replicates; SignatureAnalyzer found a false positive signature (S7 Fig) in 17 of 20 replicates.

Next, we examined a more compressive setting of nine true signatures with varying signature activities (details in Methods; S8 Fig and Table C in S1 Table). All methods correctly identified six common signatures (SBS1, 2, 3, 5, 13, 18) in 20 of 20 replicates with the cosine similarity threshold 0.8; SUITOR and SignatureAnalyzer detected two rare signatures (SBS8,41) in more replicates than the other two methods (Fig 3A). None of the methods identified the extremely rare and flat signature SBS40 (present in one tumor only). Interestingly, there existed one replicate where two detected *de novo* signatures were most similar to SBS5 (cosine similarity 0.97 and 0.91, respectively) for SUITOR. It indicates that two flat signatures (likely SBS5 and SBS40) were detected in this replicate, which is consistent with the simulation design; indeed, the signature with cosine similarity 0.91 to SBS5 was also similar to SBS40 with cosine similarity 0.83. In addition, signeR found two flat signatures in two replicates. Both flat signatures were most similar to signature SBS5 followed by SBS40. Moreover, signeR found two SBS2 signatures in 3 replicates and two SBS13 signatures in 2 replicates, suggesting that signeR would occasionally overidentify APOBEC signatures (SBS2 and SBS13). Notably, all methods detected no other signatures besides the nine true signatures. We measured the cosine similarity between detected signatures and true ones (S9A Fig) averaged over 20 replicates. The cosine similarities of five common signatures (SBS1, 2, 3, 5, 18) were all over 90% for 4 methods; the cosine similarities of SBS13 were slightly lower than 90% for SignatureAnalyzer and signeR and higher than 90% for the other two methods; as expected, the cosine similarities of two rare signatures (SBS8,41) were much lower than other signatures (except cosine similarities of SBS41 by SUITOR and SignatureAnalyzer). As a result, the increasement of cosine similarity threshold from 0.8 to 0.9 would impact the frequency of signatures SBS8, 13, 41 to be detected across 20 replicates but not other signatures (Table D in S1 Table).

Finally, we investigated the setting when part of somatic mutations was called by mistake, which likely occurs in practice due to sequencing and/or calling errors. We simulated a mutation type matrix for 300 tumors of eight signatures which cover six major substitution types. In addition, we added mutation counts caused by errors for each tumor generated from a uniform distribution with various error levels (rounded to the nearest integer, details in Methods). We repeated this simulation by 20 times. Notably, SUITOR consistently detect eight signatures regardless of error levels (Fig 3B) with the cosine similarity threshold 0.8; in contrast, although SBS9 was detected by SigProfilerExtractor (5 over 20 replicates), SignatureAnalyzer (20 over 20 replicates), and signeR (1 over 20 replicates) when there are no mutation calling errors, it was rarely detected when mutation calling errors present; for example, when the error level is 1.2, all three methods could not detect SBS9. Besides SBS9, signature SBS39 was missed by

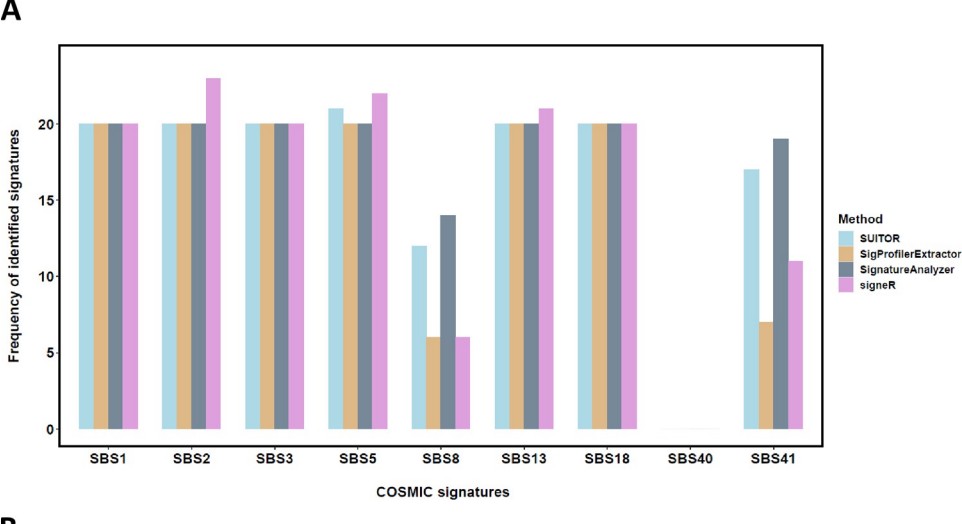

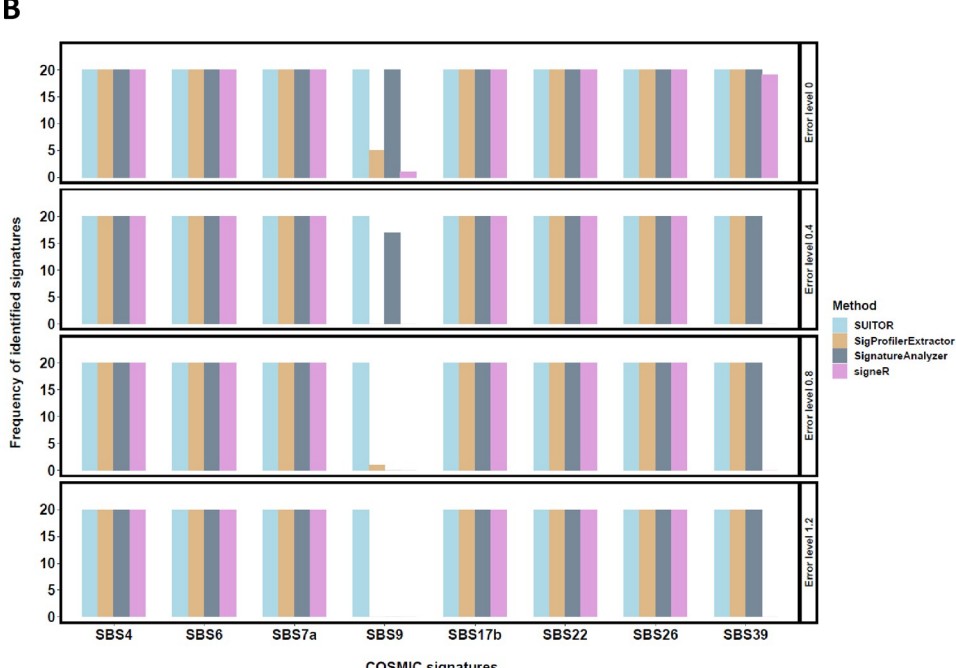

**Fig 3.** *In silico* **evaluation of SUITOR and other methods.** a) The number of replicates in which a given signature (of total 9 signatures) will be detected by each method; the rare signatures SBS8 and SBS41 were discovered in 12, 6, 14 and 6 replicates, and 17, 7, 19 and 11 replicates over total 20 replicates by SUITOR, SigProfilerExtractor, SignatureAnalyzer and signeR, respectively. b) The number of replicates in which a given signature (of total 8 signatures) will be detected by each method under various error levels. Error level equaling to zero means no somatic mutations called by errors.

signeR when there existed error mutations. Compared to the given signature profiles, cosine similarities of signatures detected by SUITOR were higher than 90% for SBS4, 6, 7a, 17b, 22, 26 and 39 and slightly lower than 90% for SBS9 (S9B Fig). Consequently, if the cosine similarity threshold is increased to 0.9, only the frequency of signature SBS9 to be detected across 20 replicates would be reduced for all methods (Table E in S1 Table), and SUITOR could still detect SBS9 in more replicates than other methods. For this simulation involved mutations called by errors, we further evaluated the performance of CV2K and SparseSignatures. When

there were no mutation calling errors, CV2K was able to detect true 8 signatures among 19 of 20 replicates; however, its performance declined with increasing error levels; when the error level is at 1.2, CV2K detected 8 signatures in 6 of 20 replicates and detected 7 signatures in other 14 replicates (Table F in S1 Table). SparseSignatures tended to find more signatures even without mutation calling errors; it detected more than 12 signatures (as many as 20 signatures for one replicate) in 9 of 20 replicates; when the error level was 1.2, SparseSignatures identified the true 8 signatures in only 2 replicates (Table F in S1 Table).

Together, the simulation studies suggest that SUITOR is able to find both common and rare signatures while well controlling the rate of false positives, outperforming other methods.

## Detection of pan-cancer mutational signatures

We tested the four methods in whole-genome sequencing (WGS) data of 2,540 tumors across 22 cancer types from the Pan-Cancer Analysis of Whole Genomes (PCAWG) study [1] (details in Methods).

First, we extracted *de novo* mutation signatures one cancer type at a time for eight cancer types, each with at least 100 tumors. Unlike in *in vitro* or *in silico* studies, the true signatures were unknown here. Nevertheless, we could evaluate if the signatures detected in part of the dataset predict mutation counts in the remaining part well. Specifically, the mutation type matrix is separated into training, validation and testing sets, the last of which is used to evaluate the performance of the selected number of signatures (details in Methods). Among the four methods, SUITOR clearly attained the smallest prediction errors across eight cancer types (Fig 4A). It suggests that the existing tools are not designed for dealing with missing data and hence cannot conduct cross-validation, which motivates us to develop SUITOR. In addition, we compared signature profiles extracted by each method, using all counts in the mutation type matrix (i.e., without the split of the mutation type matrix for SigProfilerExtractor, SignatureAnalyzer and signeR); most signatures found by SUITOR were highly similar to the COSMIC signatures [7] (with cosine similarity > 0.8, Fig 4B) and frequently detected by the other methods (Fig 4C). In contrast, SignatureAnalyzer identified more *de novo* signatures, some of which were not matched to any COSMIC signatures (Fig 4B).

Next, we extracted *de novo* signatures combining WGS data from 2,540 tumors (pan-cancer analysis). SUITOR found 22 signatures, eighteen of which could be matched to the COSMIC signatures (Fig 4B). These signatures had the smallest prediction error (Fig 4A), compared to signatures detected by the other methods: SUITOR 36,017; SigProfilerExtractor 255,111 (>7 times SUITOR's prediction error); SignatureAnalyzer 402,409 (>11 times SUITOR's prediction error); and signeR 193,191 (>5 times SUITOR's prediction error). As expected, the signatures commonly found in multiple cancer types (e.g., SBS1, SBS2 and SBS13) could be identified when combining all cancer types together, while signatures specific to a single cancer type (e.g., SBS24 specific to liver cancer) were absent in the combined signature analysis (Fig 4C). When we clustered 2,540 tumors based on signature activities **H** estimated by SUITOR, four clusters emerged in the t-SNE plot (Fig 5A). Liver tumors formed a distinct cluster, possibly due to its specific signature SBS24 caused by aflatoxin exposure; the remaining clusters included: i) a subset of lymphomas; ii) the majority of kidney tumors; iii) all remaining tumors. Similarly, we clustered the same 2,540 tumors based on signature activities **H** estimated by the other methods; liver tumors were separated from the other tumors across all of the t-SNE plots generated by the other methods (Fig 5B–5D); however, kidney tumors and lymphoma were mixed with other cancer types in the t-SNE plot from SignatureAnalyzer.

In addition, we applied CV2K and SparseSignatures to select the number of signatures. The numbers selected by CV2K were comparable to ones by SUITOR (e.g, CV2K: 8 signatures vs

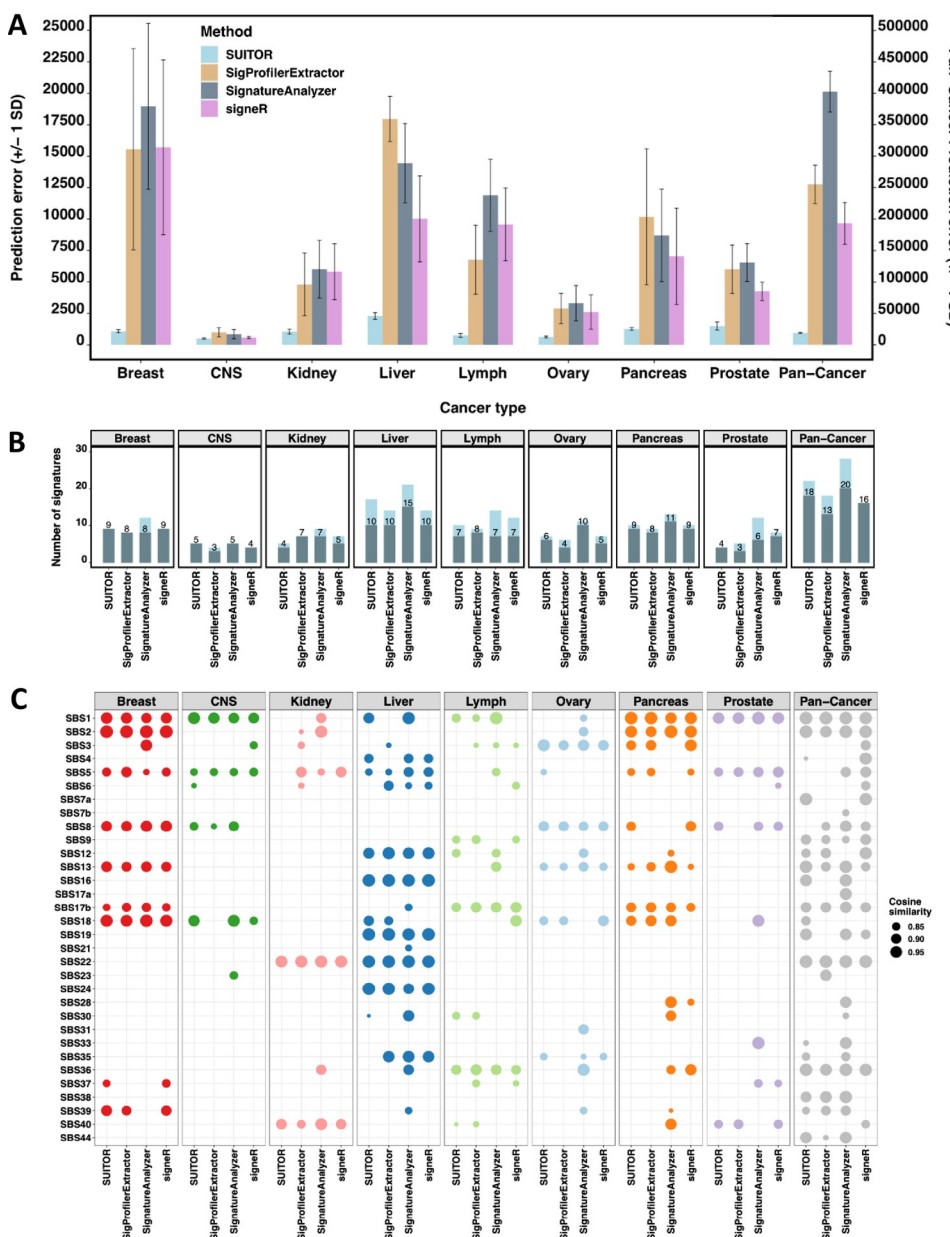

**Fig 4. Mutational signature results of eight cancer studies of PCAWG.** a) The prediction errors of SUITOR, SigProfilerExtractor, SignatureAnalyzer and signeR for eight cancer types (scale on the left-side Y axis) and for 2,540 tumors across 22 cancer types together, namely Pan-Cancer (scale on the right-side Y axis). SD: standard deviation. b) The number of signatures identified by each method. The shaded bars and the numbers above indicate the number of signatures whose profiles could be matched to COSMIC profiles (with cosine similarity > 0.8). c) The cosine similarities between *de novo* signatures and COSMIC signatures. Only the matched pairs are shown (with cosine similarity > 0.8). The higher the cosine similarity the better match to a COSMIC signature profile. The cosine similarity equivalent to one denotes a perfect match.

SUITOR: 9 signatures for breast cancer; CV2K: 4 signatures vs SUITOR: 4 signatures for prostate cancer; Table G in S1 Table). Since CV2K does not infer signatures profiles, we were unable to evaluate their similarities to COSMIC signatures. SparseSignatures tended to detect slightly fewer signatures than other methods (Table G in S1 Table); since SparseSignatures

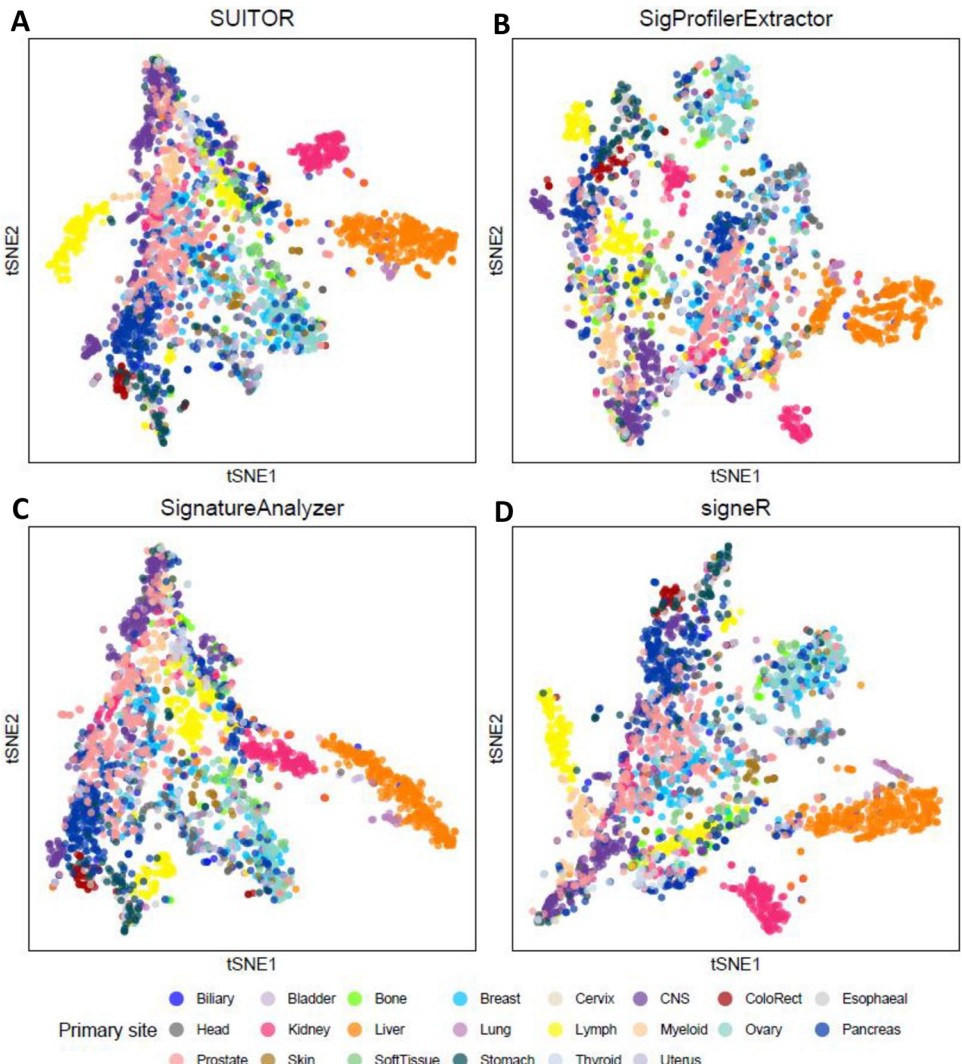

**Fig 5. The t-SNE visualization of clustering patterns for 2,540 tumors across 22 cancer types based on signature activities.** Signature activities are estimated by four methods: (a) SUITOR, (b) SigProfilerExtractor, (c) SignatureAnalyzer and (d) signeR. Each dot represents a tumor and is colored by the cancer type.

imposed the sparsity assumption, its inferred profiles were spiky and less similar to the COS-MIC signatures (Table H in S1 Table).

## External validation of breast cancer mutational signatures

We validated the nine PCAWG breast cancer signatures (based on 194 breast tumors) using an independent WGS set of 440 breast tumors of the Sanger breast cancer (BRCA) study from the same ethnicity [2]. SUITOR (Fig 6A and Table I in S1 Table) and signeR (Table J in S1 Table) identified nine breast cancer signatures in PCAWG and validated eight in the Sanger BRCA study (with cosine similarity > 0.8); SigProfilerExtractor identified eight signatures and confirmed seven (Table K in S1 Table); SignatureAnalyzer identified twelve and validated eight (Table L in S1 Table). Overall, six signatures (SBS1,2,8,13,17b and 18) were identified in both studies by all methods, while the flat featureless signature SBS5 could not be validated by any method.

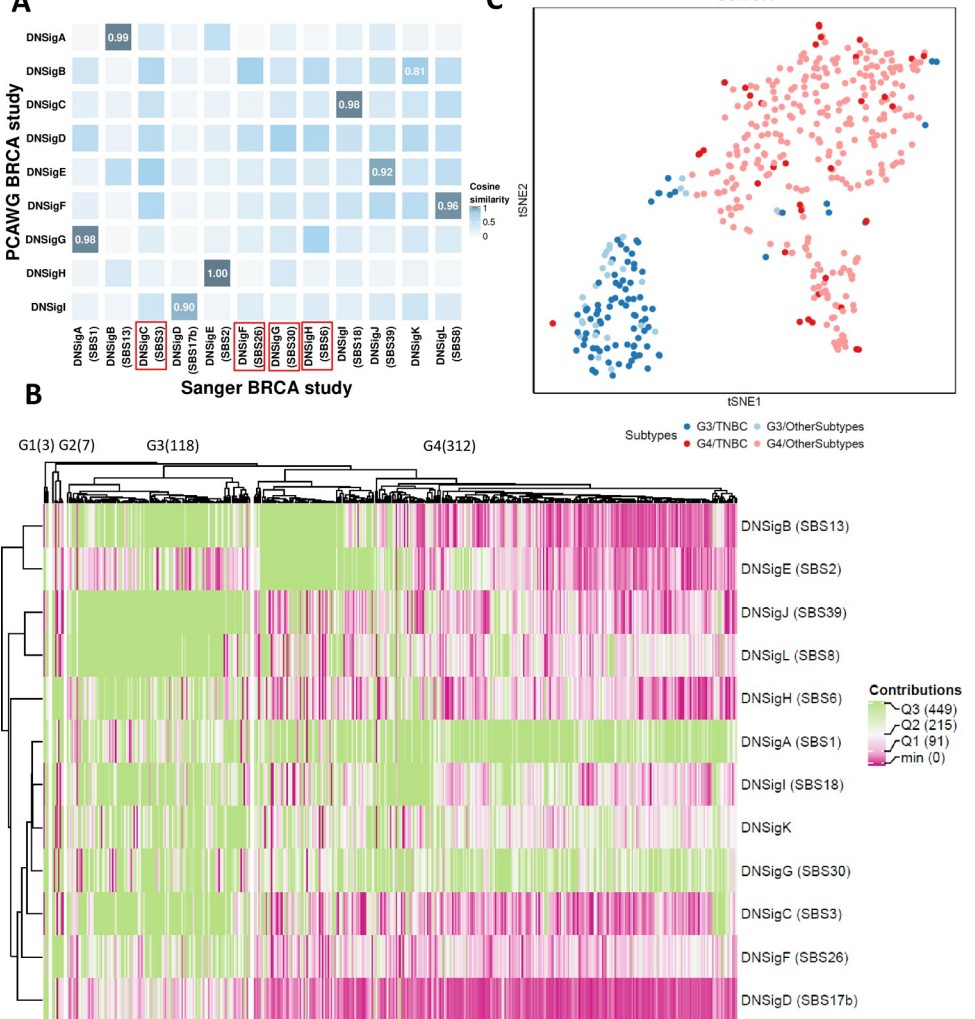

**Fig 6. The results of Sanger breast cancer study by SUITOR.** a) The heatmap of cosine similarity between *de novo* signatures detected in the PCAWG breast cancer (BRCA) study and *de novo* signatures in the Sanger BRCA study. The signatures of the Sanger BRCA study are annotated by COSMIC signatures (if cosine similarities > 0.8) among parentheses. For example, DNSigA (SBS1) refers to *de novo* signature A (DNSigA) being annotated by COSMIC signature SBS1. The four signatures which were detected in the Sanger BRCA study only were annotated by the red rectangular frames around the names of signature. b) The heatmap of signature activities estimated by SUITOR with hierarchical clustering in the Sanger BRCA study. The number of tumors included in each signature cluster is shown among parentheses on the top of the heatmap. Q1, Q2, Q3: the 1st, 2nd and 3rd quantiles of signature activities. c) The t-SNE visualization of tumors in G3 and G4 signature clusters, color-coded by signature clusters and molecular subtypes. TNBC: triple negative breast cancer.

Besides validating eight PCAWG breast cancer signatures, SUITOR found four additional signatures in the Sanger BRCA study (Fig 6A and Table M in S1 Table). Two of them were highly similar to COSMIC signatures SBS26 and SBS30 (cosine similarity > 0.93) and were identified by the other three methods as well. The other two (similar to SBS3 and SBS6) were also detected by SignatureAnalyzer and/or signeR. Similarly, three other methods found a few more signatures as well (Tables N, O, and P in S1 Table). These findings suggest that there likely exist additional signatures in the Sanger BRCA study that are missed in the PCAWG breast cancer study because of either the larger sample size or specific operative mutational processes (e.g., BRCA1 mutation carriers with signature SBS3) in Sanger BRCA study.

Finally, as an example of clinical utility of the signatures estimated by SUITOR, we stratified the 440 breast tumors of the Sanger BRCA study using the signature activities. Four signature clusters were found; two dominant clusters (G3 and G4) included overall 430 tumors (Fig 6B). Compared to the G4 subgroup, the G3 subgroup showed significantly higher activities of the nine signatures (S10 Fig and Table Q in S1 Table) and significantly lower activities of the *de novo* signature A (similar to COSMIC signature SBS1 associated with aging). We found a number of clinical factors associated with the subgroups G3 and G4, including age at diagnosis and tumor grade (Table R in S1 Table). The singular most important associated factor was the molecular subtype: triple negative breast cancers were significantly enriched in the subgroup G3 (Fig 6C; odds ratio = 25.1, P-value $< 2.2 \times 10^{-16}$, two-sided Fisher's exact test).

## Other applications

Besides selecting the number of signatures of single base substitution, SUITOR could be used to select the number of signatures of other genomic alterations in tumors, including double base substitutions, small insertion and deletions (INDELs), and structure variations. For example, SUITOR could detect four *de novo* INDEL signatures in ovarian adenocarcinoma (n = 113 tumors). Three of them were similar to the COSMIC INDEL signatures: ID1 (associated with replication slippage, cosine similarity = 0.97; Table S in S1 Table), ID6 (associated with defective homologous recombination-based DNA damage repair, cosine similarity = 0.95) and ID8 (associate with DNA double-strand breaks repair by non-homologous end joining, cosine similarity = 0.88). Although SUITOR is primarily designed for human tumors with whole genome sequencing, it can be applicated to select the number of signatures in other model organisms. As an example, we applied SUITOR to an experimental study on *C. elegans* [47]. Since mutation counts of *C. elegans* are very low (partially due to its smaller genome size around 101 Mb) relative to the mutation counts of human tumors based on WGS (human genome size around 3,000 Mb), we combined 100 *C. elegans* together to increase the mutation counts as a group (leading to 28 groups for 2717 *C. elegans*). SUITOR could detect five *de novo* SBS signatures. Compared to the signatures reported in the original *C. elegans* study [47], two *de novo* SBS signatures are highly similar to signatures of alkylating agents dimethyl sulfate (DMS, cosine similarity = 0.94, Table T in S1 Table) and ethyl methanesulfonate (EMS, cosine similarity = 1.00), respectively; and another two *de novo* SBS signatures are similar to signatures of DNA repair deficiencies due to genes *SLX1* (cosine similarity = 0.89) and *MLH1* (cosine similarity = 0.87), respectively.

For analyzing PCAWG study and Sanger breast cancer study, we excluded hypermutated tumors with mutation burden more than 10 mutations/Mb. The eight cancer types we analyzed had just few hypermutated tumors each. We further analyzed a couple of cancer types with more prevalent hypermutated tumors (colorectal adenocarcinomas n = 60, esophageal adenocarcinomas n = 98, Head and neck squamous cell carcinoma n = 57). SUITOR would find less signatures when we included hypermutated tumors (Table U in S1 Table). For example, only SBS6 (associated with defective DNA mismatch repair) and SBS10a (associated with polymerase epsilon exonuclease domain mutations) could be detected for colorectal adenocarcinoma when including hypermutated tumors (17 over 60 tumors). It suggests the inclusion of hypermutated tumors would likely hinder the detection of other signatures by SUITOR, as least when the number of tumors is small.

## Discussion

It's crucial to select the correct number of *de novo* mutational signatures for a cancer genomics study. Here we present SUITOR that selects the number of signatures through cross-validation

to minimize the prediction error in the validation set. We have shown how SUITOR outperforms common existing methods most of the time. *In vitro* studies show that SUITOR is capable of retrieving the correct number and profiles of both endogenous and exogenous signatures, allowing the correct stratification of tumor subclones exposed to distinct mutagens. *In silico* simulation studies show that SUITOR can detect common signatures in all replicates and rare signatures (as low as 1%) in the majority of replicates. Applications to eight *in vivo* PCAWG cancer types show that SUITOR discovers signatures which achieve the lowest prediction errors in the testing sets. Most of these signatures are confirmed by other methods and matched to the COSMIC signatures. All except one signature found in PCAWG BRCA study were validated in the independent Sanger BRCA study. The activities of signatures selected by SUITOR in the Sanger BRCA study are dominated by two clusters, driven by the molecular/histological subtypes.

In this paper, we used 10-fold cross validation, which is recommended as a good compromise for the bias-variance trade-off regarding the choice of k in K-fold cross validation [48,49]. In addition, we have tried 20-fold cross validation (i.e., 5% of data as validation data) for PCAWG studies, which led to the same number of signatures (results not shown).

In summary, SUITOR has shown to perform better than other commonly used methods in revealing mutational signatures, the "footprints" engraved in the cancer genomes by operative mutational processes with potentially important etiological or therapeutic implications.

## Methods

### The probabilistic non-negative matrix factorization (NMF) model

Given the number of signatures $r$ to be extracted, NMF factorizes the mutation type matrix into two non-negative matrices: the matrix of signatures $\mathbf{W}$ of size $96 \times r$ and the matrix of activities/exposures $\mathbf{H}$ of size $r \times N$ such that $\mathbf{V} \approx \mathbf{WH}$. Each column of $\mathbf{W}$ forms a signature profile with the elements summed to 1, showing how 96 mutation types comprise a signature profile; each column of $\mathbf{H}$ contains activities of $r$ signatures, reflecting how intense $r$ mutational signatures are in a tumor. To estimate $\mathbf{W}$ and $\mathbf{H}$, it is common to minimize the generalized Kullback-Leibler (KL) divergence

$$D_{KL}(\mathbf{V}|\mathbf{WH}) = \sum_{p=1}^{96} \sum_{n=1}^{N} \left\{ v_{pn} \log\left( v_{pn} \Big/ \sum_{j=1}^{r} w_{pj} h_{jn} \right) + \sum_{j=1}^{r} w_{pj} h_{jn} - v_{pn} \right\},$$

subject to $w_{pj} \geq 0$ and $h_{jn} \geq 0$ with $1 \leq j \leq r$, $1 \leq n \leq N$ and $1 \leq p \leq 96$. Lowercase letters, $v_{pn}$, $w_{pj}$ and $h_{jn}$, denote elements of the corresponding matrices, $\mathbf{V}$, $\mathbf{W}$ and $\mathbf{H}$, respectively. NMF can also be solved with other objective functions such as Frobenius norm or more general $\beta$-divergence, depending on the applications [34].

Notably, minimization of the generalized KL divergence is equivalent to maximize a likelihood function of a probabilistic NMF model [44,50]. Indeed, for a Poisson NMF model, $v_{pn}$ of the $n$th tumor and $p$th mutation type is assumed to be independently distributed, following a Poisson distribution with mean $\sum_{j=1}^{r} w_{pj} h_{jn}$. The log-likelihood of Poisson NMF model can be written as $\log\{\Pr(\mathbf{V}|\mathbf{WH})\} = -D_{KL}(\mathbf{V}|\mathbf{WH}) + C$ with a constant $C$. Therefore, minimizing generalized KL divergence $D_{KL}(\mathbf{V}|\mathbf{WH})$ is equivalent to maximizing the log-likelihood $\log\{\Pr(\mathbf{V}|\mathbf{WH})\}$. In addition, it could be shown that the multiplicative update algorithm [51], which is commonly used to minimize the generalized KL divergence, is equivalent to an expectation/conditional maximization (ECM) algorithm [40] for the Poisson NMF model (Supplementary Note 2 in S1 Text). These two equivalences are used to develop SUITOR.

## Unsupervised cross-validation for mutational signature analysis

SUITOR aims to select the optimal number of signatures which minimizes the prediction error in the validation set through cross-validation. Here, we describe the steps to create the validation set and the related challenges.

For a $K$-fold cross-validation, we divide the type matrix $\mathbf{V}$ into K parts where the Poisson NMF model is fitted on $K-1$ parts as the training set, and the fitted model is validated on the remaining one part as the validation set. The cross-validation is carried out K times with each part served as a validation set once, using the balanced separation [52,53] detailed as follows. In the $k^{th}$ fold ($1 \leq k \leq K$) of a balanced separation, a set of mutation counts $\{v_{pn}|p = (n \bmod 10) + (k-1) + aK, a = 1,2,\ldots\}$ are held out for the $n^{th}$ tumor as validation data, where $a$ is restricted such that $1 \leq p \leq 96$. For example, $(1, 11, \cdots, 91)^{st}$ mutation types of the first tumor are held out in the $1^{st}$ fold, $(2, 12, \cdots, 92)^{nd}$ mutation types in the $2^{nd}$ fold and so on. Note that the balanced separation keeps equal number of retained mutation types for each tumor in the training set, which is computationally more stable than randomly splitting $\mathbf{V}$ into the training and validation sets. The latter may randomly remove a large number of mutation types for a tumor.

As validation data are held out, missing data emerge in the training set, the reason for which existing methods of NMF fail. To address this challenge, we extended the Poisson NMF model and propose an expectation/conditional maximization (ECM) algorithm to incorporate the missing data.

## Expectation/conditional maximization (ECM) algorithm of SUITOR

Let $\mathcal{S}$ be the set of indices of mutation type matrix $\mathbf{V}$ such that $\mathcal{S} = \{(n,p)|1 \leq n \leq \mathrm{N} \text{ and } 1 \leq p \leq 96\}$. For a $K$-fold cross-validation, $\mathcal{S}$ would be divided into $K$ disjoint sets $\mathcal{S}_1, \cdots, \mathcal{S}_K$. For the $k^{th}$ fold, $\mathbf{V}_k^L = \{v_{pn}|(n,p) \in \mathcal{S}_k\}$ denotes the validation set and $\mathbf{V}_k^T = \{v_{pn}|(n,p) \in \mathcal{S} \setminus \mathcal{S}_k\}$ the training set, where $\mathcal{S} \setminus \mathcal{S}_k$ represents the indices of $\mathbf{V}$ excluding ones in $\mathcal{S}_k$. Mutation counts in $\mathcal{S}_k$ will be removed from $\mathbf{V}$ and denote as missing data $\mathbf{M}_k = \{m_{pn}|(n,p) \in \mathcal{S}_k\}$. The $v_{pn}$ in $\mathbf{V}_k^T$ and $m_{pn}$ in $\mathbf{M}_k$ are assumed to be independently distributed as a Poisson distribution with the mean $\sum_{j=1}^r w_{pj} h_{jn}$. By the scheme of balanced separation, $\mathbf{V}_k^L$ is missing completely at random (MCAR), since $\mathbf{V}_k^L$ is removed from $\mathbf{V}$, independent of values of $\mathbf{V}_k^T$ and $\mathbf{V}_k^L$. MCAR enables us to propose an ECM algorithm to incorporate missing data and obtain unbiased estimates [46] of $\mathbf{W}$ and $\mathbf{H}$.

Next, we outline the ECM algorithm in the following iterative steps (details in Supplementary Note 3 in S1 Text).

1. Initial step: choose initial values of $\mathbf{M}_k$ and set initial parameters $\mathbf{W^0}$ and $\mathbf{H^0}$.

2. E-step: given the observed data $\mathbf{V}_k^T$ and the parameters $\mathbf{W^t}$ and $\mathbf{H^t}$ of the previous step $t$, the ECM algorithm calculates the conditional expectation of complete likelihood

$$Q(\mathbf{W}, \mathbf{H}|\mathbf{W^t}, \mathbf{H^t}) = E[\log\{\Pr(\mathbf{V}_k^T, \mathbf{M}_k|\mathbf{WH})\}|\mathbf{W^t}, \mathbf{H^t}] = -D_{KL}(\mathbf{V}^*|\mathbf{WH}) + C^*,$$

where $C^*$ is a constant independent of $\mathbf{W}$ and $\mathbf{H}$, $v_{pn}^*$ the elements of $\mathbf{V}^*$ as $v_{pn}^* = v_{pn}$ for $(n,p) \in \mathcal{S} \setminus \mathcal{S}_k$ and $v_{pn}^* = E[m_{pn}] = \sum_{j=1}^r w_{pj}^t h_{jn}^t$ for $(n,p) \in \mathcal{S}_k$.

3. CM1-step: update parameters $\mathbf{W^{t+1}}$ by maximizing $Q(\mathbf{W}, \mathbf{H^t}|\mathbf{W^t}, \mathbf{H^t})$ with respect to $\mathbf{W}$.

4. CM2-step: update parameters $\mathbf{H^{t+1}}$ by maximizing $Q(\mathbf{W^{t+1}}, \mathbf{H}|\mathbf{W^t}, \mathbf{H^t})$ with respect to $\mathbf{H}$.

5. Iterate steps 2 to 4 until convergence.

In the initial step, we use the median of mutation counts in $\mathbf{V}_k^T$ per each mutation type as initial values for $\mathbf{M}_k$; other more complicated methods of specifying initial values, such as nearest neighbors, lead to similar results (results not shown). We are aware that the ECM algorithm possibly converges to a local saddle point. To overcome it, we try 300 random initial values $\mathbf{W^0}$ and $\mathbf{H^0}$, which leads to 300 pairs of $\hat{W}_i$ and $\hat{H}_i$, the estimates of $\mathbf{W}$ and $\mathbf{H}$ for the $i$th initial value, $i = 1,2,\ldots,300$. The final reported $\hat{W}$ and $\hat{H}$ are the $\hat{W}_i$ and $\hat{H}_i$ which maximize the $\log\{\Pr(\mathbf{V}_k^T|\hat{W}_l, \hat{H}_l)\}$ among all $\hat{W}_l's$ and $\hat{H}_l's$.

## Selecting number of signatures by SUITOR

For a given number of signatures $r$, we first evaluate the prediction error, i.e., the disparity between the observed validation data $\mathbf{V}_k^L$ and the predicted ones $\hat{M}_k = \mathbf{E}[\mathbf{M}_k] = \{(\hat{\mathbf{W}}\hat{\mathbf{H}})_{pn}|(n,p) \in \mathcal{S}_k\}$, for the $k^{th}$ fold, $k = 1,2,\ldots,K$,

$$ERR_{r,k} \equiv -\log\{\Pr(\mathbf{V}_k^L\hat{M}_k)\} = D_{KL}(\mathbf{V}_k^L|\hat{M}_k) - C^*.$$

We then evaluate overall prediction error, $ERR_r = \sum_{k=1}^K ERR_{r,k}$, across K folds. Since the term $C^*$ is unrelated to $\hat{\mathbf{W}}$ and $\hat{\mathbf{H}}$, it is dropped. Finally, we select the number of signatures $r^*$ which minimizes $ERR_r$ over a range of numbers of signatures $1 \leq r \leq R$.

## Extracting signature profiles and estimating activities of signatures

Once the optimal number of signatures $r^*$ is determined by SUITOR, we extract mutational signature profiles $\mathbf{W}$ and estimating signature activities $\mathbf{H}$, via maximizing $\log\{\Pr(\mathbf{V}|\mathbf{WH})\}$ with the fixed rank $r^*$. Similar to the ECM algorithm in SUITOR, we evaluate multiple initial values and use $\hat{\mathbf{W}}$ and $\hat{\mathbf{H}}$ which maximizes $\log\{\Pr(\mathbf{V}|\hat{\mathbf{W}}\hat{\mathbf{H}})\}$ to relieve local optima problem.

## Parameters used for SUITOR and other mutational signatures analysis tools

The parameters used for SUITOR and other mutational signatures analysis tools are listed as follows.

SUITOR: minimum rank: 1; maximum rank: 10; number of folds: 10; EM algorithm stopping tolerance: 1e-5; maximum number of iterations in EM algorithm: 2000; number of seeds:300

SigProfilerExtractor: sig.sigProfilerExtractor("matrix", "Output_folder_name", data, startProcess = 1, endProcess = 15, totalIterations = 100, cpu = 36).

SigneR: signeR(M = t(input), nlim = c(1,15), try_all = TRUE),

where try_all = TRUE means it evaluate BIC for rank in nlim = c(1, 15).

SignatureAnalyzer: the default parameters.

CV2K: the default parameters.

SparseSignatures: the default parameters.

## *In vitro* studies

The datasets of two *in vitro* studies were downloaded from http://medgen.medschl.cam.ac.uk/serena-nik-zainal/. The details of study design and implementation were described previously [8,11]. In these *in vitro* studies, the endogenous and exogenous mutational signatures were experimentally generated *in vitro* and hence the true number of signatures and profiles are known.

We created the mutation type matrix for both studies and applied SUITOR, SigProfilerExtractor, SignatureAnalyzer and signeR. We chose SigProfilerExtractor and SignatureAnalyzer, since they have been applied to a number of studies [13,14,28,54–56], and signeR [27] because it utilizes Bayesian information criterion (BIC), a popular model selection criterion for supervised learning. For SUITOR, we used 10-fold cross-validation with 90% of counts in mutation type matrix as the training set and the remaining 10% as the validation set. In contrast, the whole mutation type matrix $V$ was analyzed by SigProfilerExtractor, SignatureAnalyzer and signeR, respectively under the default setting.

The first study induced endogenous mutational signatures by CRISPR-Cas9-mediated knockouts of DNA repair genes in an isogenic human cell line. First, we focused on the *MSH6* knockout-induced single base substitution signature, which is characterized by C>T and T>C single base substitutions (~148 substitutions per cell division). We evaluated whether the four methods were able to retrieve the background signature and the *MSH6* knockout-induced signatures. Next, we analyzed the gene-knockout studies with no induced signatures (for genes *CHK2*, *NEIL1*, *NUDT1*, *POLB*, *POLE* and *POLM*), to evaluate whether the four methods would find false positive signatures in addition to the background signature.

In the second study, exogenous mutational signatures were created by environmental or therapeutic mutagens. We selected 324 subclones (including 35 control subclones) of human-induced pluripotent stem cell (iPSC) lines, for which the mutations were measured by whole-genome sequencing (WGS). While controls are not exposed to mutagens, each subclone is exposed to one of 79 mutagens, including simulated solar radiation (SSR), dibenzo[a,l]pyrene (DBP) and alkylating agent therapy temozolomide (TMZ). SSR recapitulates the UV-associated signatures and DBP is a potent carcinogen of the polycyclic aromatic hydrocarbons (PAHs) produced when coal, crude oil, or gasoline is burned. The stable mutational signatures were experimentally identified for 28 mutagens. Note that these 28 signature profiles identified by experiments are not distinct. Hierarchical clustering of 28 signature profiles indicated signature profiles are clustered together with 13 clusters based on cosine similarity > 0.8 between profiles (S11A Fig); consensus clustering showed that 13 clusters of signature profiles demonstrated the stable clustering (consensus matrix in S11B Fig).

For each method, we checked if the correct number of signatures were attained with its impacts on the downstream analyses. Specifically, we investigated whether the retrieved *de novo* signature profiles were highly similar to the true signature profiles. We further explored if signature contributes could separate subclones exposed to the distinctive mutagens, visualized by t-distributed stochastic neighbor embedding (t-SNE [57]).

### *In silico* simulation design of one signature

We simulated a mutation type matrix for 500 tumors and analyzed it by the four methods, each repeated 20 times. We used the signature profile of SBS8 as the true signature profile for $W$ and generated the activity vector $H$ from a uniform distribution within the range [20000, 40000]. SBS8 is dominated by C>A and T>A mutations but not as flat as signature SBS3 or SBS5 and not as spiky as signature SBS1 or SBS2. Then the mutation type $V$ was generated by a Poisson distribution with mean $WH$.

### *In silico* simulation design of nine signatures

We simulated signatures mimic to the ones observed in the Pan-Cancer Analysis of Whole Genomes (PCAWG) breast cancer study [1]. The nine signatures identified by SUITOR show various signature activities and signature profiles; some signatures contribute to all tumors (e.g., SBS1 and SBS5, present in 100% of tumors; Table C in S1 Table) while others contribute

to a few tumors (e.g., SBS41, present in 6% of tumors) and even to one (SBS40) or two tumors (SBS8); some signature profiles are spiky (e.g., SBS1 and SBS2/13; S8 Fig) while others are relatively flat (e.g., SBS5).

Specifically, we generated mutation type matrices similar to $\mathbf{V}^{BR}$, the mutation type matrix of PCAWG breast cancer study. $\mathbf{V}^{BR}$ is approximated by $\mathbf{W}^{BR}\mathbf{H}^{BR}$, for which $\mathbf{W}^{BR}$ contains the COSMIC signature profiles and $\mathbf{H}^{BR}$ is the corresponding signature activity matrix (downloaded from https://www.synapse.org/#!Synapse:syn11738669). We removed signatures with zero activities to all tumors and chose 9 signatures to compose the signature profile matrix $\mathbf{W}^{BR}$ of size 96×9. With $\mathbf{W}^{BR}$ fixed, we took bootstrap samples $\mathbf{H}^{BR}_{boot}$ from each column of matrix $\mathbf{H}^{BR}$, and generated $\mathbf{V}^{BR}_{boot}$ which follows a Poisson distribution with mean $\mathbf{W}^{BR}\mathbf{H}^{BR}_{boot}$. Due to the dependencies between SBS1 and SBS5 as well as between SBS2 and SBS13, their activities are resampled jointly while the activities of other signatures are sampled individually. We simulated 20 mutation type matrices for 200 tumors, and each was analyzed by four methods respectively.

### *In silico* simulation of eight signatures with some mutations called by mistake

We simulated a mutation type matrix for 300 tumors with mutation calling errors. We repeated this simulation 20 times and analyzed the simulated type matrices by SUITOR, SigProfilerExtractor, SignatureAnalyzer, signeR, CV2k and sparseSignatures. We chose eight signature profiles as the true signature profiles for $\mathbf{W}$ to cover all six major substitution types: SBS4 for [C > A], SBS39 for [C > G], SBS6 and SBS7a for [C > T], SBS22 for [T > A], SBS26 for [T > C], SBS17b for [T > G] and SBS9 for [T > C] and [T > G]. The activity matrix $\mathbf{H}$ was first generated from a uniform distribution with the range [0, 100], then some randomly chosen elements in the activity matrix $\mathbf{H}$ were set as zero to mimic the real data. The true mutation type $\mathbf{V^0}$ without error was generated by a Poisson distribution with mean $\mathbf{WH}$. In addition, we added error mutation counts for each mutation type and each tumor caused by possible sequencing and/or calling errors; we chose a relatively simple noise model to imitate error mutation counts not specific to a subset of mutation types. Specifically, error mutation counts were generated independently and identically from a uniform distribution for each mutation type with the range [0, $\mathbf{a}×\mathbf{b}$] and were rounded to the nearest integer, where the error level $\mathbf{a}$ equates to 0 (i.e., no error mutation counts), 0.4, 0.8 and 1.2, and $\mathbf{b}$ is the average mutation count of each tumor (i.e., $\mathbf{b}$ being the column average of the matrix $\mathbf{V^0}$). Hence, the error mutation counts were proportional to counts of true mutation counts with varying error levels.

### *In vivo* human cancer genomics studies

We analyzed 2, 540 tumors across 22 cancer types of PCAWG [1], including 321 tumors of hepatocellular carcinoma, 286 tumors of prostate adenocarcinoma, 237 tumors of pancreatic adenocarcinoma, 194 tumors of breast adenocarcinoma, 146 tumors of central nervous system medulloblastoma, 143 tumors of renal cell carcinoma,112 tumors of ovary adenocarcinoma and 100 cases of B-cell non-Hodgkin lymphoma. Other cancer types have less than 100 tumors per cancer type. The tumors were whole genome sequenced and datasets of the somatic mutation calls were downloaded from https://www.synapse.org/#!Synapse:syn11726620, which includes single base substitution (SBS) and Insertions and Deletions (INDEL). The hypermutator tumors with SBS mutation burden more than 10 mutations/Mb were excluded [58] (e.g., we excluded 5 tumors of hepatocellular carcinoma, 4 tumors of pancreatic adenocarcinoma, 4 tumors of breast adenocarcinoma, 1 tumor of renal cell carcinoma,1 tumor of ovary

adenocarcinoma, 7 tumors of B-cell non-Hodgkin lymphoma and zero tumors for other two cancer types). For each cancer type, we applied SUITOR, SigProfilerExtractor, SignatureAnalyzer and signeR to select the number of signatures and estimate signature activities and profiles. The signatures with cosine similarity larger than 0.8 were reported. The higher the cosine similarity the better match to a COSMIC signature profile. The cosine similarity equivalent to one denotes a perfect match. Note that the cosine similarity larger than 0.9 is a more stringent cutoff and our simulation studies show few true signatures may be missed by using the cosine similarity 0.9 as the cutoff.

To compare the prediction errors, we split the mutation type matrix into a training set (90% of counts in the mutation type matrix), a validation set (5%) and a testing set (5%). For SUITOR, the training set was used to fit the probabilistic NMF model with multiple numbers of signatures and the validation set to select the number of signatures. The other methods used both the training and validation sets to select the number of signatures. Next, we compared the prediction errors of selected signatures by each method on the testing set. For SigProfilerExtractor, SignatureAnalyzer and signeR, we imputed missing training data by medians of available mutation counts per each mutational type, applied each method, predicted the testing data and calculated the prediction error as SUITOR did. For SUITOR, it could handle missing data and predict the testing data by the ECM algorithm. Note that the split of the mutation type matrix into a training set (90% of counts in the mutation type matrix), a validation set (5%) and a testing set (5%) was used for comparing the prediction errors only. For comparing the extracted profiles (Fig 4B and 4C), all counts in the mutation type matrix were used for SigProfilerExtractor, SignatureAnalyzer and signeR.

## Sanger whole genome sequencing breast cancer study

The Sanger whole genome breast cancer (BRCA) study sequenced 560 breast tumors. The somatic mutation calls files were downloaded from http://ftp.sanger.ac.uk/pub/cancer/Nik-ZainalEtAl-560BreastGenomes. Among 560 breast tumors, 110 tumors were included in PCAWG and hence excluded from this validation study. Ten hypermutator tumors were also excluded. We applied SUITOR, SigProfilerExtractor, SignatureAnalyzer and signeR to a) select the number of signatures and estimate signature activities and profiles; b) compare the signatures with ones detected in the PCAWG breast cancer study; and c) investigate if additional signatures are found in Sanger whole genome breast cancer study. In addition, we stratified the tumors based on mutation activities and associated the signature clusters with epidemiological or clinical characteristics.

## Supporting information

**S1 Fig. The number of signatures selected for the knockout study of DNA repair gene *MSH6*.** Each plot shows how SigProfilerExtractor (A), SignatureAnalyzer (B) and signeR (C) select the optimal number of signatures respectively.
(TIF)

**S2 Fig. The signatures detected in knockout studies of six DNA repair genes.** A) The plots of criteria to select the optimal number of signatures by SUITOR, SigProfilerExtractor, signeR and SignatureAnalyzer (in clockwise order). B) The profiles of signatures discovered by each method.
(TIF)

**S3 Fig. Signature profiles identified experimentally from the *in vitro* study of environmental or therapeutic mutagens.** The signatures include 28 mutagen-induced signatures and a

background signature existing in control samples and all mutagen treated samples. ENU: N-ethyl-N-nitrosourea; DBP+S9: dibenzo[a,l]pyrene mixed with S9 rodent liver-derived metabolic enzyme; PhiP+S9: 2-amino-1-methyl-6-phenylimidazo[4,5-b]pyridine mixed with S9 rodent liver-derived metabolic enzyme; MNU: N-methyl-N-nitrosourea; DBPDE: dibenzo[a,l] pyrene diol-epoxide; MX: 3-chloro-4-(dichloromethyl)-5-hydroxy- 2(5H)-furanone; AAI: aristolochic acid I; 1,2-DMH+S9: 1,2-dimethylhydrazine mixed with S9 rodent liver-derived metabolic enzyme; 1,8-DNP: 1,8-Dinitropyrene; DBA+S9: dibenz[a,h]anthracene mixed with S9 rodent liver-derived metabolic enzyme; 3-NBA: 3-nitrobenzanthrone; DBADE: dibenz[a,h] anthracene diol-epoxide; 1,6-DNP: 1,6-Dinitropyrene; DES: diethyl sulfate; DBAC: dibenz[a,j] acridine; 5-Methylchrysene+S9: 5-Methylchrysene mixed with S9 rodent liver-derived metabolic enzyme; SSR: simulated solar radiation; BaP+S9: benzo[a]pyrene mixed with S9 rodent liver-derived metabolic enzyme; 6-Nitrochrysene+S9: 6-Nitrochrysene mixed with S9 rodent liver-derived metabolic enzyme; BPDE: benzo[a]pyrene-7,8-dihydrodiol-9,10-epoxide.
(TIF)

**S4 Fig. The number of signatures selected from the *in vitro* study of environmental or therapeutic mutagens by A) SigProfilerExtractor, B) SignatureAnalyzer and C) signeR.** The numbers of detected signatures include one background signature and additional mutagen-induced mutational signatures.
(TIF)

**S5 Fig. Clusters of subclones of *in vitro* study of exogenous mutagens, visualized by t-SNE for SUITOR, SigProfilerExtractor, SignatureAnalyzer and signeR.** Each dot represents a subclone, colored by the mutagen treatment.
(TIF)

**S6 Fig.** The number of signatures selected for *in silico* simulation studies with one signature by SUITOR (A), SigProfilerExtractor (B), SignatureAnalyzer (C) and signeR (D) for one replicate as an illustration.
(TIF)

**S7 Fig. The signature profiles of the true mutational signature and the ones discovered by each method for a replicate in *in silico* simulation studies with one signature.**
(TIF)

**S8 Fig. The profiles of nine signatures pre-specified in the *in silico* simulation studies.** Some signature profiles are spiky (e.g., SBS1 and SBS2/13), while others are relatively flat (e.g., SBS5).
(TIF)

**S9 Fig. The cosine similarities over 20 replicates of SUITOR, SigProfilerExtractor, SignatureAnalyzer and signeR for *in silico* simulation (A) with nine COSIMIC signatures and (B) with eight COSIMIC signatures added by sequencing errors.** The length of the solid line represents the standard deviation.
(TIF)

**S10 Fig. The boxplots of mutation contributions of 12 signatures for signature clusters G3 and G4 in Sanger breast cancer study.** The signatures of the Sanger BRCA study are annotated by COSMIC signatures (if cosine similarities > 0.8) among parentheses. For example, DNSigA (SBS1) refers to de novo signature A (DNSigA) being annotated by COSMIC signature SBS1.
(TIF)

**S11 Fig.** (A) The dendrogram of hierarchical clustering of 28 mutagens with complete linkage method. The height of the branches represents the dissimilarity defined as 1 minus cosine similarity between two signature profiles. The dashed red line corresponds to 0.8 cosine similarity which suggests 13 clusters. (B) The heatmap of the cluster consensus matrix for k = 13. Elements of the consensus matrix show the proportions of concordant mutagen pairs over resampled data: white (0%) indicates never clustered together and dark blue (100%) shows clustered together always.
(TIF)

**S1 Text. Supplementary notes.** Supplementary note 1 in S1 Text: on Bayesian information criteria (BIC) for mutational signature analysis. Supplementary note 2 in S1 Text: on equivalence between multiplicative update algorithm of NMF and exception/conditional maximization (ECM) algorithm for a Poisson NMF model. Supplementary note 3 in S1 Text: expectation/conditional maximization (ECM) algorithm.
(DOCX)

**S1 Table. Supplementary tables. Table A in S1 Table:** The cosine similarities between the *de novo* signatures and the true *in vitro* signatures. *De novo* signatures are discovered by SUITOR, SUITOR, SigProfilerExtractor, SignatureAnalyzer and signeR. Cosine similarity is used to compare them with true *in vitro* signatures: the background signature and signature induced by MSH6 gene knockout. Cosine similarity ranges from 0 to 1, with a cosine of 1 indicating a perfect match. **Table B in S1 Table:** The cosine similarities between the signatures discovered by SUITOR, SigProfilerExtractor, SignatureAnalyzer, signeR and the true mutagen-induced signatures. For each *de novo* signature, the largest cosine similarities are highlighted in red, indicating the strongest similarity to the corresponding *in vitro* signatures. ENU: N-ethyl-N-nitrosourea; DES: diethyl sulfate; 1,2-DMH+S9: 1,2-dimethylhydrazine mixed with S9 rodent liver-derived metabolic enzyme; TMZ: Temozolomide; SSR: simulated solar radiation; 6-NC: 6-Nitrochrysene; AAI: aristolochic acid I. **Table C in S1 Table:** The percentage of nonzeros, mean and standard deviation (SD) of signature contributions in *in silico* simulation studies of nine signatures. **Table D in S1 Table:** The selected number of signatures out of 20 in *in silico* simulations of nine signatures for SUITOR, SigProfilerExtractor, SignatureAnalyzer, signeR. The column with #(CS > 0.8) indicates the number of signatures based on the threshold 0.8, while the column with #(CS>0.9) indicates the number of signatures based on the threshold 0.9. **Table E in S1 Table:** The selected number of signatures out of 20 in *in silico* simulations with sequencing errors for SUITOR, SigProfilerExtractor, SignatureAnalyzer, signeR. The column with #(CS > 0.8) indicates the number of signatures based on the threshold 0.8, while the column with #(CS>0.9) indicates the number of signatures based on the threshold 0.9. Sequencing errors per each tumor were generated from a uniform distribution within the range [0, a×b], where the letter a denotes ErrorLevel (0 means no sequencing errors) and b is the average mutation count of each tumor. The frequencies of signature SBS9 are highlighted in red because the frequency of signature SBS9 to be detected across 20 replicates would be reduced for all methods if the cosine similarity threshold is increased to 0.9. **Table F in S1 Table:** The selected number of signatures out of 20 in *in silico* simulations with sequencing errors for SparseSignatures and CV2K. Sequencing errors per each tumor were generated from a uniform distribution within the range [0, a×b], where the a denotes ErrorLevel (0 means no sequencing errors) and b is the average mutation count of each tumor. **Table G in S1 Table:** The number of signatures identified by SparseSignatures and CV2K for eight cancer types of PCAWG studies, entire PCAWG data and Sanger Breast cancer study. The column with CS* > 0.8 indicates the number of signatures whose largest cosine similarity is greater than 0.8 for SparseSignatures method. **Table H in S1 Table:** The cosine similarities between *de*

*novo* signatures discovered by SparseSignatures in *in vivo* studies. We annotated them with COSIMC signature having the largest cosine similarity. COSMIC signatures and cosine similarities are highlighted in red if it is larger than 0.8. Duplicated signatures with repsect to the highest cosine similarity are marked with an asterisk. **Table I in S1 Table:** The cosine similarity between *de novo* signatures discovered by SUITOR in PCAWG breast cancer study and *de novo* signatures discovered by SUITOR in Sanger breast cancer study. For each column, the largest cosine similarity is highlighted in red if it is larger than 0.8 or in blue otherwise, corresponding to the COSMIC signature most similar to a given PCAWG or Sanger signature. **Table J in S1 Table:** The cosine similarity between *de novo* signatures discovered by signeR in PCAWG breast cancer study and *de novo* signatures discovered by signeR in Sanger breast cancer study. For each column, the largest cosine similarity is highlighted in red if it is larger than 0.8 or in blue otherwise, corresponding to the COSMIC signature most similar to a given PCAWG or Sanger signature. **Table K in S1 Table:** The cosine similarity between *de novo* signatures discovered by SigProfilerExtractor in PCAWG breast cancer study and de novo signatures discovered by SigProfilerExtractor in Sanger breast cancer study. For each column, the largest cosine similarity is highlighted in red if it is larger than 0.8 or in blue otherwise, corresponding to the COSMIC signature most similar to a given PCAWG or Sanger signature. **Table L in S1 Table:** The cosine similarity between *de novo* signatures discovered by SignatureAnalyzer in PCAWG breast cancer study and *de novo* signatures discovered by SignatureAnalyzer in Sanger breast cancer study. For each column, the largest cosine similarity is highlighted in red if it is larger than 0.8 or in blue otherwise, corresponding to the COSMIC signature most similar to a given PCAWG or Sanger signature. **Table M in S1 Table:** The cosine similarity between *de novo* signatures discovered by SUITOR in PCAWG breast cancer study and in Sanger breast cancer study with COSMIC signature profiles. For each column, the largest cosine similarity is highlighted in red if it is larger than 0.8 or in blue otherwise, corresponding to the COSMIC signature most similar to a given PCAWG or Sanger signature. **Table N in S1 Table:** The cosine similarity between *de novo* signatures discovered by SigProfilerExtractor in PCAWG breast cancer study and in Sanger breast cancer study with COSMIC signature profiles. For each column, the largest cosine similarity is highlighted in red if it is larger than 0.8 or in blue otherwise, corresponding to the COSMIC signature most similar to a given PCAWG or Sanger signature. **Table O in S1 Table:** The cosine similarity between *de novo* signatures discovered by SignatureAnalyzer in PCAWG breast cancer study and in Sanger breast cancer study with COSMIC signature profiles. For each column, the largest cosine similarity is highlighted in red if it is larger than 0.8 or in blue otherwise, corresponding to the COSMIC signature most similar to a given PCAWG or Sanger signature. **Table P in S1 Table:** The cosine similarity between *de novo* signatures discovered by signeR in PCAWG breast cancer study and in Sanger breast cancer study with COSMIC signature profiles. For each column, the largest cosine similarity is highlighted in red if it is larger than 0.8 or in blue otherwise, corresponding to the COSMIC signature most similar to a given PCAWG or Sanger signature. **Table Q in S1 Table:** The mean and standard deviation (SD) of signature contributions by signature subtype G3 and G4. **Table R in S1 Table:** The clinical and pathological factors associated with the subgroups G3 and G4. a) For continuous factors; b) for discrete factors. **Table S in S1 Table:** The cosine similarities between *de novo* INDEL signatures in ovarian adenocarcinoma by SUITOR and COSIMC INDEL signatures. **Table T in S1 Table:** The cosine similarity between *de novo* signatures discovered by SUITOR and the signatures reported in the original *C. elegans* study (Volkova et al. 2020). COSMIC signatures and cosine similarities are highlighted in red if it is larger than 0.8. **Table U in S1 Table:** The cosine similarity between *de novo* signatures discovered by SUITOR in PCAWG colorectal, esophageal, and head SCC cancer studies. COSMIC signatures and cosine similarities are highlighted in

red if it is larger than 0.8. The numbers in parentheses are the number of tumors for each study.
(XLSX)

**S1 Data. Information for software and dataset webpages.**
(DOCX)

**S2 Data. The input and output matrices of signature profiles (W) and activities (H) for the SUITOR as well as the benchmarking methods (SigProfilerExtractor, SignatureAnalyzer, and signeR) as results of *in vitro* and the Pan-Cancer Analysis of Whole Genomes (PCAWG) studies.**
(ZIP)

## Acknowledgments

This study utilized the high-performance computational capabilities of the Biowulf Linux cluster at the National Institutes of Health, Bethesda, MD: https://biowulf.nih.gov. We would like to thank Bill Wheeler (Information Management Services) for computation support and Dr. Paul Albert (Biostatistics Branch of DCEG) and Dr. Ludmil Alexandrov (University of California San Diego) for helpful comments. We appreciate the help from authors of CV2K and SparseSignatures.

## Author Contributions

**Conceptualization:** Xiaohong R. Yang, Jianxin Shi, Maria Teresa Landi, Bin Zhu.

**Data curation:** Donghyuk Lee, Difei Wang.

**Formal analysis:** Donghyuk Lee, Difei Wang.

**Funding acquisition:** Bin Zhu.

**Investigation:** Donghyuk Lee, Bin Zhu.

**Methodology:** Donghyuk Lee, Bin Zhu.

**Project administration:** Bin Zhu.

**Resources:** Xiaohong R. Yang, Jianxin Shi, Maria Teresa Landi, Bin Zhu.

**Software:** Donghyuk Lee, Difei Wang.

**Supervision:** Jianxin Shi, Maria Teresa Landi, Bin Zhu.

**Validation:** Donghyuk Lee, Bin Zhu.

**Visualization:** Donghyuk Lee, Difei Wang.

**Writing – original draft:** Donghyuk Lee, Bin Zhu.

**Writing – review & editing:** Donghyuk Lee, Xiaohong R. Yang, Jianxin Shi, Maria Teresa Landi, Bin Zhu.

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
