## [Decision Letter · Decision Letter 0]

17 Aug 2021

Dear Dr. Zhu,

Thank you very much for submitting your manuscript "SUITOR: selecting the number of mutational signatures through cross-validation" for consideration at PLOS Computational Biology.

As with all papers reviewed by the journal, your manuscript was reviewed by members of the editorial board and by three independent reviewers. In light of the reviews (below this email), we would like to invite the resubmission of a significantly-revised version that takes into account the reviewers' comments.

We cannot make any decision about publication until we have seen the revised manuscript and your response to the reviewers' comments. Your revised manuscript is also likely to be sent to reviewers for further evaluation.

Sincerely,

Anna R Panchenko

Associate Editor

PLOS Computational Biology

Jian Ma

Deputy Editor

PLOS Computational Biology

Reviewer's Responses to Questions

**Comments to the Authors:**

Reviewer #1: Comments for the manuscript draft and supplementary figures are uploaded as attachments. Comments for the supplementary tables are described below.

Supplementary table comments:

In the description of Supplementary Table 3, "signarues" should be corrected to "signatures".

In Supplementary Table 4, row "SBS40", column "nonzeros SD", please ensure that this NA value is correct.

For supplementary tables 5, 6, 7, 8, 10, 11 and 12, please ensure that the significance of highlighted values is explicitly stated in the table caption.

For supplementary tables 5, 6, 7, 8, 9, 10, 11 and 12, please ensure that the significance of color coding for highlighted values (red versus blue) is explicitly stated in the table caption.

For supplementary tables 5, 6, 8, 9, 10, 11 and 12, please ensure that the highlighted values are consistently displayed in either a bold typeface or non-bold typeface.

Reviewer #2: Comments for Authors

In this study, the authors describe the development of a new bioinformatic tool intended to deal with one of the main issues during mutational signature analysis, which is the correct selection of the number of signatures.

In my opinion, the authors did a great job in addressing one of the key points in this relatively new type of genomic analysis, since many of the previously published tools have failed in solving it. Actually, there are a substantial number of R-based tools for mutational signatures analysis recently published, including MutationalPatterns (doi: 10.1186/s13073-018-0539-0), MutSpec (doi: 10.1186/s12859-016-1011-z), SomaticSignatures (doi: 10.1093/bioinformatics/btv408) or SignatureToolsLib (doi: 10.1038/s43018-020-0027-5), that do not provide automatic selection of the number of signatures. Therefore, these tools leave one of the key parameters of the analysis directly on the users’ hands. However, many times users are not experts on the computational engines behind the signature extraction process. In this regard, SUITOR appears as a very interesting option to work alongside these other tools, in order to correctly select the number of signatures that better represent the data.

However, the authors have not explored this possibility in the manuscript, focusing on benchmarking their tool against three other tools that do include automatic selection of the number of signatures. Although the approach for the selection of the number of signatures provides a new perspective with the insightful use of cross-validation, the tool suffers a lack of completeness in comparison to the other three tools benchmarked in the manuscript, SigProfilerExtractor, SignatureAnalyzer and SigneR. SUITOR does not include almost any user guide, explanation of the input parameters and output files/variables generated, plotting capabilities, possibility of extracting signature and activity matrices, comparison to existing reference signatures (e.g., COSMIC database) or algorithm tuning. Even though the reported benchmarking results are promising, the authors should consider including some of these features to facilitate the usage of SUITOR. Currently, I was not able to perform a simple analysis with the provided test data and obtaining the extracted mutational signatures. I correctly obtained the suggested number of signatures, but not the actual signature profiles.

Please find below some more detailed comments:

Major comments

1. The main concern regarding the presented bioinformatic tool, SUITOR, is that is unclear if it can be used for a complete mutational signature extraction analysis or only for the selection of the number of signatures. According to the provided benchmarking figures and tables, as well as the description of the ECM algorithm, it seems that SUITOR can extract mutational signatures by itself. However, the R code following the instructions provided does not allow to generate the matrices of signatures and activities, giving only the suggested number of signatures as result.

2. The output matrices for the benchmarking are not available (only the cosine similarities in the supplementary tables), which prevents reproducing the reported extraction results for SUITOR or the other three tools.

3. The rationale for the selection of the benchmarking datasets is not provided in some cases, e.g., selection of SBS8 for the in silico simulations with one signature, or exclusion of hypermutated tumors with >10 mutations/Mb in PCAWG and breast cancer data analysis. Actually, hypermutated tumors are particularly important in mutational signature analysis since they can lead to the extraction of false positive signatures (doi: 10.1038/s41586-020-1943-3). Therefore, their influence on SUITOR performance should be assessed.

In addition, the authors should make a clear distinction between benchmarking, which is only possible in silico using simulated data, and exploratory analysis using real data (either in vitro or in vivo). Benchmarking is only possible in simulations because ground truth signatures are needed for comparison, and these are not available in real data.

Also, ground truth signature/s are not available for the first in silico analysis performed in the manuscript (lines 436-440) since the input matrix was constructed using random numbers, instead of generating a random ground truth signature/s. Moreover, the authors claimed that the signature extracted from this random-generated input matrix is similar to SBS3 (line 215), which is probably related to the fact that SBS3 is a flat-like signature. However, SBS3 has been previously associated with defective homologous recombination-based DNA damage repair and germline mutations in BRCA1 and BRCA2 (doi: 10.1016/j.cell.2012.04.024, 10.1038/nature17676, 10.1038/nm.4292, 10.1038/onc.2016.243 and others).

4. The authors should clarify why the PCAWG benchmarking on the three additional tools was performed on 90% of the counts in the mutation catalog matrix (lines 482-491) and not on 100% as it was done in the in vitro benchmarking (lines 410-412). This cross-validation approach may have biased the benchmarking results since the other tools are not designed for dealing with missing data. The authors used the medians of available mutation counts per mutational context to fill the missing data, which may have disrupted the original input matrix and the whole extraction process.

5. As previously mentioned, the GitHub README file is missing different sections, in particular a detailed explanation of the different input parameters, as well as the output variables and objects. A more detailed vignette would also be useful apart from the html file provided (https://github.com/binzhulab/SUITOR/blob/master/SUITOR.html), in a similar way as the SigneR authors presented for their tool (https://www.bioconductor.org/packages/release/bioc/vignettes/signeR/inst/doc/signeR-vignette.html). An example of a comprehensive mutational signature extraction analysis should be included in this vignette in order to be reproducible by potential users.

6. In a SUITOR test following the code provided in the html file and using the provided test data but fixing the print parameter on print=3, it is concerning to observe that some iterations of the EM algorithm did not converge. It is not clear in the manuscript how many iterations are used as maximum for the multiplicative updates/ECM algorithm, as well as how many NMF/ECM replicates are considered.

Minor comments

1. The authors should clarify why 0.8 was used as the threshold for cosine similarity matching instead of the 0.9 commonly used in the prior literature (doi: 10.1038/s41586-020-1943-3).

2. The inclusion of the developed package in one of the main repositories for R packages (R CRAN or Bioconductor) is recommended since this has also been done by many of the other R-based mutational signature analysis tools already available in the field, including MutationalPatterns, SomaticSignatures or SigneR.

3. The background section is missing the newest collection of reference mutational signatures in the COSMIC database, corresponding to version 3.2 from March 2021 (https://cancer.sanger.ac.uk/signatures/). The authors should also mention in the introduction the other mutational types where mutational signatures have been extracted, including indels, doublet base substitutions (reference signatures also present in COSMIC), structural variants (doi: 10.1038/nature17676 among others) and copy number variants (doi: 10.1038/s41588-018-0179-8, 10.1101/2021.04.30.441940 and others).

4. The association of NTHL1 germline mutations with a particular mutational signature (SBS30) in more than 10 different cancer types should also be included in the background section in line 34 apart from the association with breast cancer (doi: 10.1016/j.ccell.2018.12.011).

5. The claim about the method used by SomaticSignatures to select the number of signatures (lines 44-46) is inaccurate, since this tool does not have an automatic method for this selection. The authors correctly pointed out that the authors of SomaticSignatures suggested using the residual sum of squares and the explained variance. However, this is only a suggestion, and the user is required to manually choose the number of signatures to use in the extraction based on a set of plots depicting the mentioned metrics as well as some additional ones.

6. The language used for the first explanation of the specific cross-validation scheme developed for the mutational signature analysis problem in lines 77-83 is unclear, making it difficult to follow.

7. In the mathematical notation of the method, W and H matrices are switched with regard to the original implementation of the mutational signatures framework (doi: 10.1016/j.celrep.2012.12.008 and 10.1038/s41586-020-1943-3). W was originally defined as the matrix of signatures (rows = number of mutational contexts & columns = signatures) and H as the matrix of activities/exposures (rows = signatures & columns = samples). In the manuscript, the authors changed the notation and transposed the matrices, which is mathematically correct but not in agreement with previous literature.

8. The authors should clarify which data was used for generating the clustering presented in figure 5 and in the results section (lines 254-256).

9. The authors should provide some evidence that SUITOR is able to handle the selection of the number of signatures arising from other genomic alterations (as they claimed in the discussion section, lines 312-314) since only data from single base substitutions (SBS96 context) is presented in the manuscript.

10. The parameters used for the different tools are missing in the methods section, in particular, the number of NMF/ECM runs performed and the maximum number of iterations per run in case that convergence is not reached, including the ones used for SUITOR (lines 393-397).

11. SigProfilerExtractor should be used as the name of the corresponding tool instead of sigProfiler, according to the information present in the GitHub provided in line 510 and also by the authors in their preprint for that particular tool (doi: 10.1101/2020.12.13.422570).

12. Since the specific version number is not available, an approximate date of download should be included for SignatureAnalyzer in line 512.

13. The authors should consider moving some technical details of the results-overview section (lines 97-156) to the methods section.

14. A citation to the COSMIC database (e.g. doi: 10.1093/nar/gky1015) should be included in the first mention of the COSMIC reference mutational signatures, as well as in line 32 when mentioning the current reference set of mutational signatures. The URL https://cancer.sanger.ac.uk/signatures/ may also be included.

15. A citation to a review about the several mutational signature analysis tools in the literature should be added in lines 42-43. Some examples are doi: 10.1093/bib/bbx082, 10.1016/j.mam.2019.05.002 or 10.1371/journal.pone.0221235. In addition, the authors should mention the two main types of analysis that have been addressed by prior studies: signature extraction and signature refitting based on reference mutational signatures. Whereas the authors have focused their manuscript on the extraction process, it is important to mention the availability of the refitting analysis, with potential applicability in clinical practice (as discussed here, doi: 10.1186/s13059-016-0893-4).

Reviewer #3: This paper addresses a key challenge in mutational signatures analysis of cancer genomes that is determining the correct number of distinct mutational processes/signatures operating on studied samples, i.e. the rank of factorization. The problem is important, especially that it significantly affects the downstream analyses. The method is based on the cross-validation method used for model selection. The authors evaluate their method on simulated and in vitro data, as well as real cancer samples. They compare it with existing methods developed for signature inference (not specifically for the identification of number of signatures).

Overall the manuscript is well written and presents an interesting approach, but it lacks clarity in some parts. The significance of the study lies in stating an important practical problem directly and proposing a method to solve it. However, it seems that the authors are not aware of existing approaches for this problem that use cross-validation as well. CV2K method (Gilad et al, Mach. Learn. Sci. Technol., 2020) uses a similar cross-validation schema (although random, not balanced separation as the authors) combined with sBCD NMF. SparseSignatures (Lal et al., PLoS Comp. Biol. 2021; available in BioRxiv and Bioconductor since 2018), a framework to extract signatures, uses cross-validation to identify the number of signatures. These two methods seem to be more appropriate for benchmarking the method, as they use a similar cross-validation idea (both methods) or were developed specifically for the same problem (CV2K).

Please do not change existing terminology used in signature analysis literature; see the seminal paper by Alexandrov et al. (Cell Reports, 2013) for reference. It can lead to misunderstanding and confusion. For example, a mutational catalog of a cancer genome is defined by Alexandrov et al. as a vector containing the number of somatic mutations of a genome defined over a finite alphabet of mutation types. Thus, “A[C>G]G” is a mutation type not a mutation catalog; and there are 96 mutation types (or categories, features) not catalogs.

The authors use two in vitro studies to benchmark their approach. In both of these studies there is only one perturbation per experiment (gene knockout or cell exogenous exposure) leading to at most one mutational signature beyond the background. However, there is an experimental dataset available that has both gene knockouts and genotoxins, see the paper by Volkova et al. (Nature Communications, 2020). The authors could evaluate their approach on this dataset if possible (e.g. samples that have high number of mutations).

Is there a particular reason that the two in vitro datasets were treated in a different way? The samples from Kucab et al. were all analysed together (put in a single V matrix), while the samples from Zou et al were split into two analyses (MSH6 knockout and the rest). All methods should be able to find two signatures in Zou et al even when all samples were analysed together. Only MSH6 KO samples have two signatures present while all other samples should have only one signature (background) with non-zero activity.

For the second in vivo dataset (Kucab et al), all methods found a different number of signatures. However, how many signatures should be expected for this dataset? Have you tried to see how many profiles of these 28 mutagens are similar to each other? If you assume 0.8 cosine similarity (as you do for signature comparison), how many clusters of mutagen profiles would you have? Moreover, have you inspected the exposure matrix inferred by different methods? Each studied sample comes from a single mutagen treatment, so its profile should be affected by at most two signatures (background and mutagen signature). Thus, the exposure matrix (for all samples as you have in your paper) should be very sparse with at most two non-zero values in each row; more non-zero values would suggest overfitting and the number of signatures is too high and should be reduced. This could lead to a good validation of the result.

Figure 2b, top panel, is incorrect. It shows the MSH6 knockout signature not the background signature. Both top and bottom panels (Fig. 2b) show the same signature, but the top panels should show the background signature that is similar to signatures shown in Supplemental Figure 2b, or signature in Fig 3a (parental clone) in the original paper (Zou et al).

The random mutation scenario in in silico simulation study seems artificial as it is not similar to any existing COSMIC signature. The proposed association with SBS3 is weak; SBS3 is usually associated with HR deficiency in signature studies. Instead, you should expand your two other scenarios. More signatures could be tested in your design with one signature, not only SBS8. However, a design with more signatures is more realistic and corresponds to actual problems in cancer studies. Thus, you could expand the last scenario in place of the first two. It will be more informative.

In Figure 3b, some methods, including SUITOR, find the same signature more than 20 times in 20 replicates. This suggests overfitting as the same COSMIC signature is found more than once. Can it be related to a case when prediction error for a validation set exhibits a “plateau” near the minimum, i.e. similar prediction error for different number of signatures? In such a case it might be reasonable to select the lowest possible number of signatures not the number that gives the lowest error as it might be statistically impossible to differentiate these two cases.

How are the missing values selected in the initial step of the algorithm? In line 376, it is stated that you use the median of mutation counts in V (training) per each mutation catalog (meaning mutation type), i.e. median of non-missing values from a corresponding column of V. However, in the detailed description of the algorithm (Supplementary note 3), it stated that these are the medians from the corresponding row of V. Please fix this (probably in Supplementary note 3) and potentially other places where it is incorrectly stated.

The above mentioned initialization is okay for SUITOR as it tries to improve the estimates of the held-out mutation counts. However, it does not seem appropriate to fill in missing values in datasets provided to other methods (SigProfiles, SignatureAnalyser, signeR) as you do in the in vivo (PCAWG) benchmarking. Missing mutation count data is not a problem in mutational signature analysis, so these methods were not designed to deal with it. SUITOR imputes them because it is part of the cross-validation schema it uses. This leads to unfair comparison between SUITOR and other methods. It would be fair if the authors provide to the other methods the values imputed by SUITOR to fill in the held-out mutation counts. This way the comparison between methods would assess the methods’ ability to infer the number of signatures and their profiles, not the other methods’ inability to correctly infer the missing values and signature profiles based on incorrect input data. As it is, the imputed values are not even normalized to a sample's overall mutation counts; for each mutation type, all missing values are replaced by the same value (column median). Thus, SUITOR has an advantage when compared with other methods.

Minor comments:

- Use correct names for algorithms (upper/lowercase), i.e. SigProfiler, SignatureAnalyser, signeR, consistently.

- line 52: “elaborated” > “are elaborated”

- line 125: “been” > “be”

- line 138: “The” > “the”

- line 359: fix the reference

**Have the authors made all data and (if applicable) computational code underlying the findings in their manuscript fully available?**

Reviewer #1: Yes

Reviewer #2: **No: **Signature and activity matrices arising from the benchmarking results have not been made available by the authors. They are needed for the reproducibility of the study.

Reviewer #3: Yes

PLOS authors have the option to publish the peer review history of their article (what does this mean?). If published, this will include your full peer review and any attached files.

Reviewer #1: **Yes: **Daniel Espiritu

Reviewer #2: No

Reviewer #3: No
---

## [Decision Letter · Decision Letter 1]

21 Feb 2022

Dear Dr. Zhu,

Thank you very much for submitting your manuscript "SUITOR: selecting the number of mutational signatures through cross-validation" for consideration at PLOS Computational Biology. As with all papers reviewed by the journal, your manuscript was reviewed by members of the editorial board and by several independent reviewers. Based on the reviews, we are likely to accept this manuscript for publication, providing that you modify the manuscript according to the review recommendations. Please make sure that you make all data sets available, as was requested by reviewers.

Sincerely,

Anna R Panchenko

Associate Editor

PLOS Computational Biology

Jian Ma

Deputy Editor

PLOS Computational Biology

[LINK]

Reviewer's Responses to Questions

**Comments to the Authors:**

**Reviewer #2:** I am sincerely grateful for the thorough review of the manuscript performed by the authors. In particular, they did a great job in improving the usability of the tool and providing the specific analyses suggested by the other reviewers and myself regarding different variant types, species or hypermutated tumors, increasing the quality of the manuscript and the tool. Considering the benchmarking reported, SUITOR appears as a powerful tool for the accurate selection of the number of mutational signatures in a specific cancer dataset, especially for non-hypermutated tumors. However, I have some additional comments that can be found below.

1. As I previously mentioned, the commonly used cosine similarity threshold for mutational signature analysis in previous literature is 0.9. The authors claimed that their simulation studies show minor differences between this well-established threshold and their selected one (0.8). However, they should provide some detailed metrics about the differences observed in the benchmarking if they had used 0.9 (or other intermediate value) as the threshold. For example, in the case of the supplementary table 2 (in vitro analysis), using a different cosine similarity threshold would result in SUITOR having a reduced precision/specificity compared to the other three tools, especially SigneR and SigProfilerExtractor. The selection of the cosine similarity threshold is critical considering that the average cosine similarity expected between two random nonnegative vectors simply by chance is 0.75 (doi: 10.1186/s12859-020-03772-3).

2. For the real data analyses, the authors validate their results by indicating the matching of the extracted signatures to COSMIC reference signatures. This matching by itself is not an indication that the tool is identifying real signatures in the data. For example, in the analysis of the C. elegans dataset, the authors claimed to identify a signature closely matching SBS7c. This signature, as indicated, has a specific etiology (UV light exposure) and is usually found alongside SBS7a/b in skin tumors. Therefore, it seems unlikely that this signature is present in the original dataset by itself and could potentially represent a false positive signature. In addition, the authors should compare their results with the ones from the original C. elegans study (doi: 10.1038/s41467-020-15912-7).

3. Also regarding the C. elegans dataset, the authors should clarify if they have renormalized the COSMIC reference signatures to account for the differences in the trinucleotide proportions between human and C. elegans genomes. One example of the renormalization can be observed in the COSMIC website (https://cancer.sanger.ac.uk/signatures/downloads/), where the original GRCh37-based reference signatures were translated to different reference genomes and species, according to the different trinucleotide frequencies.

4. The authors should explain why they chose to include different noise levels generated from a uniform distribution, considering the data is an integer (counts). An expansion of the methods regarding the noise model selection would also be helpful.

5. COSMIC Mutational Signatures v3.2 includes 78 single base substitution signatures and not 49, since several artifact-related signatures are also included in the reference set.

6. The results of the application of SUITOR to hypermutated tumors, indels, and the C. elegans study should be included in the results section apart from the discussion section. For the analysis of hypermutators and indel signatures, it should be indicated which datasets were used and how many samples were included per group. For indels and the C. elegans study, a supplementary table with the cosine similarities similar to the ones provided for the other datasets is missing.

7. For SignatureAnalyzer, it should be specified if the CPU or the GPU version was used.

8. The output W and H matrices are missing for the newly included datasets of PCAWG Colorectal, PCAWG Esophageal, PCAWG Head and Neck, ovarian adenocarcinoma for indel analysis, and C. elegans analysis. Also, to benefit reproducibility, the mutational matrices used as input for all benchmarking and real data analyses should be included in the supplementary dataset.

9. Typos: spacing between equal signs not consistent (e.g., lines 389-391); C. elegans should be written in italics (lines 394-397); *caner (line 407). Please check for typos in the user guide too (e.g., *SUTIOR, section 4.2).

**Reviewer #3:** The reviewer's comments were fully addressed. No more comments.

**Have the authors made all data and (if applicable) computational code underlying the findings in their manuscript fully available?**

Reviewer #2: **No: **The output W and H matrices are missing for the newly included datasets of PCAWG Colorectal, PCAWG Esophageal, PCAWG Head and Neck, ovarian adenocarcinoma for indel analysis, and C. elegans analysis. Also, to benefit reproducibility, the mutational matrices used as input for all benchmarking and real data analyses should be included in the supplementary dataset.

Reviewer #3: Yes

PLOS authors have the option to publish the peer review history of their article (what does this mean?). If published, this will include your full peer review and any attached files.

Reviewer #2: No

Reviewer #3: No

Figure Files:

Data Requirements:

Reproducibility:

References:

---

## [Editor Report · Decision Letter 2]

9 Mar 2022

Dear Dr. Zhu,

We are pleased to inform you that your manuscript 'SUITOR: selecting the number of mutational signatures through cross-validation' has been provisionally accepted for publication in PLOS Computational Biology.

Best regards,

Anna R Panchenko

Associate Editor

PLOS Computational Biology

Jian Ma

Deputy Editor

PLOS Computational Biology

---

## [Editor Report · Acceptance letter]

29 Mar 2022

PCOMPBIOL-D-21-01288R2 

SUITOR: Selecting the number of mutational signatures through cross-validation

Dear Dr Zhu,

I am pleased to inform you that your manuscript has been formally accepted for publication in PLOS Computational Biology. Your manuscript is now with our production department and you will be notified of the publication date in due course.

With kind regards,

Zsofia Freund
